# Modelling Cellular Perturbations with the Sparse Additive Mechanism Shift Variational Autoencoder

**Michael Bereket**
insitro*
mbereket@stanford.edu

**Theofanis Karaletsos**
insitro*
theofanis@karaletsos.com

## Abstract

Generative models of observations under interventions have been a vibrant topic of interest across machine learning and the sciences in recent years. For example, in drug discovery, there is a need to model the effects of diverse interventions on cells in order to characterize unknown biological mechanisms of action. We propose the Sparse Additive Mechanism Shift Variational Autoencoder, SAMS-VAE, to combine compositionality, disentanglement, and interpretability for perturbation models. SAMS-VAE models the latent state of a perturbed sample as the sum of a local latent variable capturing sample-specific variation and sparse global variables of latent intervention effects. Crucially, SAMS-VAE sparsifies these global latent variables for individual perturbations to identify disentangled, perturbation-specific latent subspaces that are flexibly composable. We evaluate SAMS-VAE both quantitatively and qualitatively on a range of tasks using two popular single cell sequencing datasets. In order to measure perturbation-specific model-properties, we also introduce a framework for evaluation of perturbation models based on average treatment effects with links to posterior predictive checks. SAMS-VAE outperforms comparable models in terms of generalization across in-distribution and out-of-distribution tasks, including a combinatorial reasoning task under resource paucity, and yields interpretable latent structures which correlate strongly to known biological mechanisms. Our results suggest SAMS-VAE is an interesting addition to the modeling toolkit for machine learning-driven scientific discovery.

## 1 Introduction

Scientific discovery often involves observation and intervention on systems with the aim of eliciting a mechanistic understanding. For example, in biology, large cellular perturbation screens with high-dimensional readouts have become increasingly popular as an approach to investigate biological mechanisms, their regulatory dependencies, and their responses to drugs. As technology enables both richer and finer grained measurements of these systems, there is an increasing need and opportunity for machine learning methods to help generate predictive insights of growing complexity.

Generative models such as variational auto-encoders (VAEs) [8] are commonly used to learn representations of complex datasets and their underlying distributions. A common goal in generative modeling is disentanglement, whereby latent structures should factorize into semantic subspaces to facilitate generalization and discovery. A desirable outcome consists of these subspaces learned by models being indicative of latent mechanisms, while sparsely varying according to the underlying latent factors of variation in the true data distribution [10]. This goal has recently been formalized under the Sparse Mechanism Shift framework [20, 9] which connects disentanglement to the causal inference field through the identification of causal graphs. Concomitantly, recent models such as the Compositional Perturbation Autoencoder [13] and SVAE+ [12] have successfully applied disentangled deep learning to scientific problems in single-cell RNA-sequencing under perturbation.

---

*Research supporting this publication conducted while authors were employed at insitro

In this work, we propose the Sparse Additive Mechanism Shift Variational Autoencoder (SAMS-VAE), a model which extends prior work by capturing interventions and their sparse effects as explicit additive latent variables. Compared to previous approaches for modeling disentanglement in VAEs applied to cellular data, our model explicitly combines sparse perturbation-specific latent effects, perturbation-independent natural variation of cells, and additive composition of perturbation effects in a joint model. We also introduce CPA-VAE, which ablates the sparsity mechanism we propose, yielding a generative model with similar assumptions as the popular perturbation model CPA. To perform approximate inference, we propose rich variational families for these models and showcase how sophisticated inference facilitates identifying predictive factors of variation. We additionally introduce a lens on evaluation of perturbation models for biology based on model-based average treatment effects and differential expression, which we link to posterior predictive checks.

In our experiments we showcase SAMS-VAE in various tasks across cellular sequencing data. We observe that SAMS-VAE achieves superior predictive capability over baselines across two popular single-cell sequencing datasets in tasks related to in-distribution- and out-of-distribution generalization, including combinatorial generalization when multiple perturbations are applied. We furthermore examine the interpretability of the model's disentangled structures and demonstrate significantly improved ability to recover factors predictive of known molecular pathways as compared to recently proposed models. Finally, we show that our best models also excel in the treatment effect estimation evaluation we propose.

## 2 The Sparse Additive Mechanism Shift Variational Autoencoder

We consider datasets $(\boldsymbol{x}_i, \boldsymbol{d}_i)_{i=1}^N$ of observations $\boldsymbol{x}_i \in \mathbb{R}^{D_x}$ and perturbation dosage vectors $\boldsymbol{d}_i \in \{0,1\}^T$, where $d_{i,j}$ is 1 if sample $i$ received perturbation $j$ and 0 otherwise. We aim to develop generative models of $p(\boldsymbol{x}|\boldsymbol{d})$, representing the distribution of features of a target system conditional on perturbations. In the following sections, we will introduce the details of our proposed modeling strategy, the Sparse Additive Mechanism Shift Variational Autoencoder (SAMS-VAE).

### 2.1 Generative model

We consider generative models with the following basic structure:

$$\boldsymbol{z}_i = \boldsymbol{z}_i^b + \boldsymbol{z}_i^p$$
$$\boldsymbol{x}_i \sim p(\boldsymbol{x}_i|\boldsymbol{z}_i; \boldsymbol{\theta})$$

$\boldsymbol{z}_i \in \mathbb{R}^{D_z}$ is the latent state embedding for sample $i$, which is modeled as the sum of a latent basal state embedding $\boldsymbol{z}_i^b \in \mathbb{R}^{D_z}$ and a latent perturbation effect embedding $\boldsymbol{z}_i^p \in \mathbb{R}^{D_z}$. Observations are then sampled from a conditional likelihood $p(\boldsymbol{x}_i|\boldsymbol{z}_i; \boldsymbol{\theta})$. In this paper, we focus on likelihoods for $p(\boldsymbol{x}_i|\boldsymbol{z}_i; \boldsymbol{\theta})$, where parameters are computed from $\boldsymbol{z}_i$ using a neural network with parameters $\boldsymbol{\theta}$.

The core modeling assumption of SAMS-VAE relates to the distribution $p(\boldsymbol{z}_i^p|\boldsymbol{d}_i)$. We propose to model perturbations as inducing sparse latent offsets that compose additively as follows:

$$\boldsymbol{z}_i^p = \sum_{t=1}^T d_{i,t}(\boldsymbol{e}_t \odot \boldsymbol{m}_t), \tag{1}$$

where $\boldsymbol{e}_t \in \mathbb{R}^{D_z}$ and $\boldsymbol{m}_t \in \{0,1\}^{D_z}$ are global latent variables that determine the latent offset due to perturbation $t$. $\boldsymbol{m}_t$ is a binary mask that performs feature selection on the latent offset $\boldsymbol{e}_t$: when $\boldsymbol{m}_t$ is sparse $\boldsymbol{e}_t \odot \boldsymbol{m}_t$ will result in a sparse offset. Importantly, global variables $\boldsymbol{e}_t$ and $\boldsymbol{m}_t$ are shared across all samples (corresponding to cells) that receive perturbation $t$.

We specify the prior distributions $p(\boldsymbol{e}_t) \sim \mathcal{N}(0, \beta I)$ and $p(\boldsymbol{m}_t) = \text{Bern}(\alpha)$ for perturbation effects, where $\alpha$ is chosen to be small to induce sparsity. While we focus on the Bernoulli prior for the mask, we also provide a Beta-Bernoulli prior in our code for mask $\boldsymbol{m}_t$ as an easy plug-in replacement. We omit additional prior assumptions regarding the structure of perturbation effects in this work. We specify the prior distribution $p(\boldsymbol{z}_i^b) \sim \mathcal{N}(0, I)$ for latent basal states.

Using this latent structure, we define the full generative model for SAMS-VAE as in Figure 5. The joint probability distribution over our observed and latent variables is defined as:

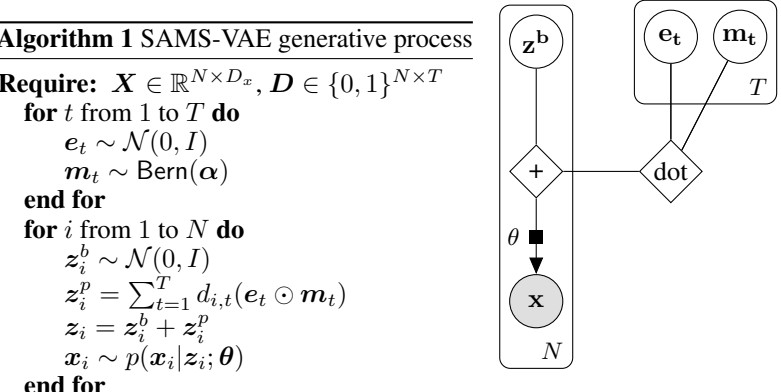

**Algorithm 1** SAMS-VAE generative process

**Require:** $\boldsymbol{X} \in \mathbb{R}^{N \times D_x}$, $\boldsymbol{D} \in \{0,1\}^{N \times T}$
    **for** $t$ from 1 to $T$ **do**
        $\boldsymbol{e}_t \sim \mathcal{N}(0, I)$
        $\boldsymbol{m}_t \sim \mathrm{Bern}(\boldsymbol{\alpha})$
    **end for**
    **for** $i$ from 1 to $N$ **do**
        $\boldsymbol{z}_i^b \sim \mathcal{N}(0, I)$
        $\boldsymbol{z}_i^p = \sum_{t=1}^{T} d_{i,t}(\boldsymbol{e}_t \odot \boldsymbol{m}_t)$
        $\boldsymbol{z}_i = \boldsymbol{z}_i^b + \boldsymbol{z}_i^p$
        $\boldsymbol{x}_i \sim p(\boldsymbol{x}_i | \boldsymbol{z}_i; \boldsymbol{\theta})$
    **end for**

Figure 1: SAMS-VAE represented as an generative process (left) and as a graphical model (right).

$$p(\boldsymbol{X}, \boldsymbol{Z}^b, \boldsymbol{M}, \boldsymbol{E} | \boldsymbol{D}; \boldsymbol{\theta}) = \left[ \prod_{t=1}^{T} p(\boldsymbol{e}_t)p(\boldsymbol{m}_t) \right] \left[ \prod_{i=1}^{N} p(\boldsymbol{z}_i^b)p(\boldsymbol{x}_i | \boldsymbol{z}_i^b, \boldsymbol{d}_i, \boldsymbol{M}, \boldsymbol{E}; \boldsymbol{\theta}) \right], \tag{2}$$

for observations $\boldsymbol{X} \in \mathbb{R}^{N \times D_x}$, perturbation dosages $\boldsymbol{D} \in \{0,1\}^{N \times D_z}$, latent basal states $\boldsymbol{Z}^b \in \mathbb{R}^{N \times D_z}$, latent perturbation embeddings $\boldsymbol{E} \in \mathbb{R}^{T \times D_z}$, and latent perturbation masks $\boldsymbol{M} \in \{0,1\}^{T \times D_z}$.

## 2.2 Likelihood choice for scRNA-Seq Data

In the previous section, we used a generic form $p(\boldsymbol{x}_i | \boldsymbol{z}_i; \boldsymbol{\theta})$ for the observation model over $\boldsymbol{x}$ to show the generality of the approach. Below, we describe the observation model we use to apply SAMS-VAE to single cell RNA-sequencing data (scRNA-seq) in more detail.

We represent scRNA-seq observations as $\boldsymbol{x}_i \in \mathbb{N}^{D_x}$, where each value $x_{i,j}$ is the number of measured transcripts in cell $i$ that correspond to gene $j$, and follow the likelihood introduced by Lopez et al. [11, 12] to model the elaborate noise structure of scRNA-seq data. An additional utility quantity *library size* $l_i$, the total number of transcripts measured in cell $i$, is included as an observed variable in the conditioning set. This is useful because the library size is largely determined by technical factors that we are not interested in modeling. The full likelihood function is then defined as follows:

$$\boldsymbol{\rho}_i = f_\theta(\boldsymbol{z}_i)$$
$$\boldsymbol{\lambda}_i \sim \Gamma(\boldsymbol{\rho}_i l_i, \theta_d)$$
$$\boldsymbol{x}_i \sim \mathrm{Poisson}(\boldsymbol{\lambda}_i),$$

where $\boldsymbol{\rho}_i \in [0,1]^{D_x}$ represents the expected frequency of each transcript in cell $i$ and is parameterized by $f_\theta$, a neural network with a softmax output. Observations are then sampled from a Gamma-Poisson distribution (equivalently, a negative binomial distribution) with mean $\boldsymbol{\rho}_i l_i \in \mathbb{R}_+^{D_x}$ and inverse dispersion $\theta_d \in \mathbb{R}_+^{D_x}$. $\theta_d$ is a learned parameter that is shared across cells.

## 2.3 Inference

We perform inference on SAMS-VAE using stochastic variational inference [4, 8] to approximate the marginal likelihood $\log p(\boldsymbol{X} | \boldsymbol{D})$ by optimizing parameters $\phi$ and $\boldsymbol{\theta}$. We do so by maximizing the evidence lower bound (ELBO) for SAMS-VAE, defined as follows:

$$\mathrm{ELBO}(\boldsymbol{\phi}, \boldsymbol{\theta}) = \mathbb{E}_{\boldsymbol{Z}^b, \boldsymbol{E}, \boldsymbol{M} \sim q(\cdot | \boldsymbol{X}, \boldsymbol{D}; \boldsymbol{\phi})} \log \frac{p(\boldsymbol{X}, \boldsymbol{Z}^b, \boldsymbol{M}, \boldsymbol{E} | \boldsymbol{D}; \boldsymbol{\theta})}{q(\boldsymbol{Z}^b, \boldsymbol{M}, \boldsymbol{E} | \boldsymbol{X}, \boldsymbol{D}; \boldsymbol{\phi})}. \tag{3}$$

A key question when performing variational inference is the choice of variational family to approximate the posterior distribution. As a baseline inference strategy, we consider the following **amortized mean-field** inference scheme for SAMS-VAE:

$$q(\boldsymbol{Z}^b, \boldsymbol{M}, \boldsymbol{E} | \boldsymbol{X}, \boldsymbol{D}; \boldsymbol{\phi}) = \left[ \prod_{t=1}^{T} q(\boldsymbol{m}_t; \boldsymbol{\phi}) q(\boldsymbol{e}_t; \boldsymbol{\phi}) \right] \left[ \prod_{i=1}^{N} q(\boldsymbol{z}_i^b | \boldsymbol{x}_i; \boldsymbol{\phi}) \right]. \tag{4}$$

We parameterize $q(\boldsymbol{m}_t; \boldsymbol{\phi}) = \text{Bern}(\hat{\boldsymbol{p}}_t)$ and $q(\boldsymbol{e}_t; \boldsymbol{\phi}) = \mathcal{N}(\hat{\boldsymbol{\mu}}_t, \hat{\boldsymbol{\sigma}}_t)$ with learnable parameters $\hat{\boldsymbol{p}}_t, \hat{\boldsymbol{\mu}}_t, \hat{\boldsymbol{\sigma}}_t$. We define $q(\boldsymbol{z}_i^b | \boldsymbol{x}_i) = \mathcal{N}(\hat{f}_{enc}(\boldsymbol{x}_i))$, where $\hat{f}_{enc}$ is a learnable neural network that predicts mean and standard deviation parameters. During training, gradients are computed for $q(\boldsymbol{m}_t; \boldsymbol{\phi})$ with a Gumbel-Softmax straight-through estimator [5].

We propose **two improvements** to the mean-field inference scheme that aim to more faithfully invert the SAMS-VAE generative model. First, we model possible correlations between sample latent basal states $\boldsymbol{z}_i^b$ and the global latent perturbation masks and embeddings (**correlated encoder**). We do so by implementing $q(\boldsymbol{z}_i^b | \boldsymbol{x}_i, \boldsymbol{d}_i, \boldsymbol{E}, \boldsymbol{M}) = \mathcal{N}(\hat{f}_{enc}([\boldsymbol{x}_i \; \boldsymbol{z}_i^p]))$ for $\boldsymbol{z}_i^p$ as defined in equation 1, where $\hat{f}_{enc}$ is a neural network that takes as input both the observations and the estimated latent perturbation effect embeddings for a given sample. Second, we model possible correlations between the latent perturbation masks and embeddings by replacing $q(\boldsymbol{e}_t)$ with $q(\boldsymbol{e}_t | \boldsymbol{m}_t)$ (**correlated embeddings**). We implement $q(\boldsymbol{e}_t | \boldsymbol{m}_t) = \mathcal{N}(\hat{f}_{emb}(\boldsymbol{m}_t, t))$ with a learnable neural network $\hat{f}_{emb}$ that predicts the embedding from a mask and a one-hot encoding of the treatment index. Applying both of these modifications, we define the **correlated variational family** for SAMS-VAE as:

$$q(\boldsymbol{Z}^b, \boldsymbol{M}, \boldsymbol{E} | \boldsymbol{X}, \boldsymbol{D}) = \left[ \prod_{t=1}^{T} q(\boldsymbol{m}_t; \boldsymbol{\phi}) q(\boldsymbol{e}_t | \boldsymbol{m}_t; \boldsymbol{\phi}) \right] \left[ \prod_{i=1}^{N} q(\boldsymbol{z}_i^b | \boldsymbol{x}_i, \boldsymbol{d}_i, \boldsymbol{E}, \boldsymbol{M}; \boldsymbol{\phi}) \right]. \tag{5}$$

This richer variational family posits a joint infinite mixture variational distribution between the global and local variables in the model to finely capture their interdependencies and we evaluate its components separately in our experiments. We elaborate on the objective per minibatch and other details in our supplemental Section A.1

## 2.4 CPA-VAE

To directly assess the effect of the sparsity inducing masks, we define an ablated model, CPA-VAE, that is identical to SAMS-VAE with all mask components fixed to 1. Thus, in contrast to equation 1, we have that $\boldsymbol{z}_i^p = \sum_{t=1}^{T} d_{i,t} \boldsymbol{e}_t$. We call this model CPA-VAE because it directly incorporates the additive latent composition assumption from CPA [13], and CPA-VAE can be thought of as an extension of CPA to a fully specified generative model. We note that CPA-VAE inherits the benefits of the inference improvements to the variational families we propose in this work and will assess the contributions of better inference and sparse masking separately.

# 3 Quantitative Evaluation of Perturbation Models

In this section we discuss two quantitative strategies to rigorously evaluate our perturbation models. First, we discuss how to estimate the marginal likelihood for our model on held-out data, a common strategy employed across generative modeling to assess density estimation. Second, we define a posterior predictive check for model predictions of average treatment effects.

## 3.1 Marginal Likelihood

We consider the marginal log likelihood of held out data under an inferred generative model, estimated via the importance weighted ELBO (IWELBO) [1], as our primary evaluation metric (a similar metric was used in Lopez et al. [12]) Specifically, we estimate $\log P(\boldsymbol{X} | \boldsymbol{D}, \boldsymbol{\theta})$ on held out data, where $\boldsymbol{\theta}$ denotes decoder parameters. Let $\boldsymbol{H} = (\boldsymbol{Z}^b, \boldsymbol{M}, \boldsymbol{E})$ represent the set of latent variables for SAMS-VAE. Then we can write the importance weighted ELBO with $K$ particles as:

$$\text{IWELBO}(\boldsymbol{X} | \boldsymbol{D}, \boldsymbol{\phi}, \boldsymbol{\theta}) = \mathbb{E}_{\boldsymbol{H}^{(1)}, \dots, \boldsymbol{H}^{(K)} \sim q(\boldsymbol{H} | \boldsymbol{X}, \boldsymbol{D}, \boldsymbol{\phi})} \log \frac{1}{K} \sum_{k=1}^{K} \frac{p(\boldsymbol{X}, \boldsymbol{H}_k | \boldsymbol{D}, \boldsymbol{\theta})}{q(\boldsymbol{H}_k | \boldsymbol{X}, \boldsymbol{D}, \boldsymbol{\phi})}.$$

The importance weighted ELBO can be used to holistically compare the generalization of generative models such as SAMS-VAE, CPA-VAE, and SVAE+. We note, however, that a marginal likelihood cannot be computed for models that are not fully specified as probabilistic models. In practice, we estimate IWELBO as follows:

$$\text{IWELBO}(\boldsymbol{X}|\boldsymbol{D},\boldsymbol{\phi},\boldsymbol{\theta}) = \mathbb{E}_{\boldsymbol{H}^{(1)},\dots,\boldsymbol{H}^{(K)}\sim q(\boldsymbol{H}|\boldsymbol{X},\boldsymbol{D},\boldsymbol{\phi})} \log \frac{1}{K} \sum_{k=1}^{K} w_k, \tag{6}$$

for

$$w_k = \left[ \frac{p(\boldsymbol{M}^{(k)},\boldsymbol{E}^{(k)}|\boldsymbol{\theta})}{q(\boldsymbol{M}^{(k)},\boldsymbol{E}^{(k)}|\boldsymbol{\phi})} \prod_{i=1}^{N} \frac{p(\boldsymbol{x}_i|\boldsymbol{z}_i^{b(k)},\boldsymbol{M}^{(k)},\boldsymbol{E}^{(k)},\boldsymbol{d}_i,\boldsymbol{\theta})p(\boldsymbol{z}_i^{b(k)})}{q(\boldsymbol{z}_i^{b(k)}|\boldsymbol{M}^{(k)},\boldsymbol{E}^{(k)},\boldsymbol{X},\boldsymbol{d}_i,\boldsymbol{\phi})} \right]. \tag{7}$$

## 3.2 A Posterior Predictive Check for Average Treatment Effects of Perturbation Models

As a second category of metrics, we consider posterior predictive checks (PPC) [2, 19]: we query test statistics of interest in the predictive distribution of learned models and compare these statistics against estimates from the data. These types of assessments can be useful when critiquing models for specific use cases, such as predicting the mean of some measurement under different perturbations. However, these assessments only characterize narrow aspects of the predictive distribution, providing a less complete assessment than the marginal likelihood.

As a test statistic for our PPC we choose the population average treatment effect of a perturbation relative to a control perturbation on each measurement $x_{i,j}$ for sample $i$ and gene $j$, given given as:

$$\text{ATE} = \mathbb{E}_{i \in \mathcal{D}}\left[ [x_{i,j}|\text{do}(\boldsymbol{d}^*)] - [x_{i,j}|\text{do}(\boldsymbol{d}_0)] \right].$$

We define the average treatment effect for SAMS-VAE as $\text{ATE}_{\text{SAMS-VAE}}(\boldsymbol{d}^*|\boldsymbol{D}_m)$ for an applied treatment $\boldsymbol{d}^*$ and conditioning data $\boldsymbol{D}_m$ (the training data) as the difference between the expected predictive value of output variable $x_{i,j}$ given a treatment $\boldsymbol{d}^*$ and the expected predictive value of $x_{i,j}$ given control treatment $\boldsymbol{d}_0$:

$$\text{ATE}_{\text{SAMS-VAE}}(\boldsymbol{d}^*|\boldsymbol{D}_m) = \mathbb{E}_{p(\boldsymbol{z}_j^b)p(\boldsymbol{E},\boldsymbol{M}|\boldsymbol{D}_m)}\left[ T_1 - T_2 \right],$$

with $T_1 := \mathbb{E}_{p(x_{i,j}|\text{do}(\boldsymbol{d}^*),\boldsymbol{z}_j^b,\boldsymbol{M},\boldsymbol{E})}[x_{i,j}]$ and $T_2 := \mathbb{E}_{p(x_{i,j}|\text{do}(\boldsymbol{d}_0),\boldsymbol{z}_j^b,\boldsymbol{M},\boldsymbol{E})}[x_{i,j}]$.

Both of the inner expectations share global and local latent variables and only differ in the treatments, while marginalizing over observation noise. We thus disentangle between noise differences caused by a treatment, since observation noise is marginalized out. We also marginalize over the prior basal state $p(\boldsymbol{z}^b)$ in the outer expectation, which simulates populations of different cells varying by natural variation. In practice we draw $K$ samples $\boldsymbol{z}_k^b \sim p(\boldsymbol{z}^b)$ and $\boldsymbol{E}_k, \boldsymbol{M}_k \sim p(\boldsymbol{E},\boldsymbol{M}|\boldsymbol{D}_m)$ for the outer expectation and evaluate the inner expectations by a small amount of $S$ samples. In cases where observations contain multiple features (i.e. genes $j$), this quantity yields a vector per feature. Using this approach we can generate perturbed cells using the dosages $\boldsymbol{d}$. We note that other models are treated equivalently when feasible by handling their global and local variables analogously.

Because we cannot directly observe counterfactuals, we must identify a related observed quantity to evaluate our model estimated average treatment effects. We reach for differential expression (DE), a commonly chosen metric to study sequencing data collected under different conditions. A key difference between differential expression and average treatment effects is differential expression's computation based on differences of population averages.

A second key difference is that the model-based ATE marginalizes out observation noise per sample, while DE cannot distinguish noise from perturbation effects. We note that differential expression as such takes the form of the following expression and is computed over a dataset $\boldsymbol{D}_\delta$ over which the expectation is computed (where $\boldsymbol{D}_\delta^{\boldsymbol{d}}$ denotes the subset of the dataset $\boldsymbol{D}_\delta$ under condition $\boldsymbol{d}$):

$$\text{DE}_{\text{Data}}(\boldsymbol{d}^*|\boldsymbol{D}_\delta) = \mathbb{E}_{x_i \sim \boldsymbol{D}_\delta^{\boldsymbol{d}^*}}[x_{i,j}|\text{do}(\boldsymbol{d}^*)] - \mathbb{E}_{x_i \sim \boldsymbol{D}_\delta^{\boldsymbol{d}_0}}[x_{i,j}|\text{do}(\boldsymbol{d}_0)].$$

To create the metric **ATE-Pearson** $r(\text{ATE}_{\text{SAMS-VAE}}(\boldsymbol{d}^*|\boldsymbol{D}_m), \text{DE}_{\text{Data}}(\boldsymbol{d}^*|\boldsymbol{D}_\delta))$ we compute the Pearson correlation coefficient $r$ between differential expression estimates from data $\text{DE}_{\text{Data}}$ and our model-based estimator $\text{ATE}_{\text{SAMS-VAE}}$ across all features (commonly genes indexed by $j$).

This also reveals the relationship to a PPC $p(r(\text{ATE}_{\text{SAMS-VAE}}(\boldsymbol{d}^*|\boldsymbol{D}_m), \text{DE}_{\text{Data}}(\boldsymbol{d}^*|\boldsymbol{D}_\delta))|\boldsymbol{D}_m, \boldsymbol{D}_\delta)$, considering ATE as the diagnostic statistic which is approximated by DE in the observed sample. The dataset $\boldsymbol{D}_m$ for conditioning or training the model and a dataset $\boldsymbol{D}_\delta$ for estimating differential expression may be the same or different, depending on the use case. We note that the utilization of a separate dataset for a PPC is unconventional, but has previously been used in HPCs [14].

## 4 Experiments

**Overview**  We compare SAMS-VAE with baseline models through a series of applications to perturb-seq datasets. Perturb-seq is a type of biological experiment in which cells are individually perturbed and subsequently profiled with single cell RNA sequencing (scRNA-seq). Single cell RNA sequencing measures the count of messenger RNA (mRNA) transcripts (also called gene expression) for thousands of genes in each cell, providing a rich, high-dimensional characterization of cellular state. Common perturbation types for perturb-seq experiments include genetic knockouts, which disable the expression of target genes through gene editing, and chemical compounds.

In our experiments, we represent perturb-seq datasets as a gene expression matrix $X \in \mathbf{N}^{N \times D_x}$ and a perturbation dosage matrix $T \in \{0, 1\}^{N \times T}$ for $N$ cells, $D_x$ gene transcripts, and $T$ perturbations. Entries $X_{i,j}$ represent the number of transcripts from gene $j$ observed in cell $i$ and entries $T_{i,k}$ represent whether cell $i$ received perturbation $k$. We compare SAMS-VAE against baseline models based on their ability to model the distribution of perturb-seq data, generalize to new perturbations in combinatorial settings, and disentangle known biological pathways in their latent variables.

**Baselines**  We consider **CPA-VAE**, **SVAE+**, and **conditional VAE** [21] as baselines. As discussed in Section 2.4, CPA-VAE can be thought of as an extension of CPA [13] to a fully specified generative model and takes advantage of our proposed correlated inference strategy. We additionally consider ablations of the correlated inference strategies for SAMS-VAE and CPA-VAE. Complete details of model choices are provided in the appendix.

**Code availability**  Our code, which includes implementations of all models and experiment configurations, is available at `https://github.com/insitro/sams-vae`.

### 4.1 Generalization under individual perturbations

**Dataset**  To assess model generalization to held out samples under individual perturbations, we analyze a subset of the genome-wide CRISPR interference (CRISPRi) perturb-seq dataset from Replogle et al. [17], which we call `replogle-filtered`. CRISPRi is a type of genetic perturbation that represses the expression of selected target genes. Following the preprocessing steps from Lopez et al. [12], `replogle-filtered` is filtered to contain perturbations that were identified as having strong effects and genes that were associated with these perturbations. We additionally include cells with non-targeting CRISPR guides to use as controls for average treatment effect prediction. All together, `replogle-filtered` contains 118,461 cells, 1,187 gene expression features per cell, and 722 unique CRISPR guides (perturbations). We randomly sample train, validation, and test splits.

| Model | Inference | Test IWELBO | Mask PW. Acc. | ATE-Pearson |
|---|---|---|---|---|
| Conditional VAE | amortized MF | $-1766.10 \pm 0.18$ | - | **0.765** |
| SVAE+ | amortized MF | $-1761.42 \pm 0.06$ | $0.78 \pm 0.04$ | 0.605 |
| CPA-VAE | amortized MF | $-1760.14 \pm 0.20$ | - | 0.523 |
| CPA-VAE | corr. $z_{basal}$ | $-1756.57 \pm 0.14$ | - | 0.571 |
| SAMS-VAE | amortized MF | $-1757.72 \pm 0.14$ | $0.68 \pm 0.09$ | 0.302 |
| SAMS-VAE | corr. $E$ | $-1758.08 \pm 0.07$ | $0.71 \pm 0.04$ | 0.319 |
| SAMS-VAE | corr. $z_{basal}$ | $-1756.40 \pm 0.06$ | $0.87 \pm 0.02$ | 0.718 |
| SAMS-VAE | corr. $z_{basal}$ and $E$ | $\mathbf{-1756.27 \pm 0.10}$ | $\mathbf{0.89 \pm 0.03}$ | **0.765** |

Table 1: Quantitative evaluation of treatment effects on Replogle filtered dataset (100 latent dimensions) using $K = 10.000$ samples. We find that inference strategies utilizing correlated variational families lead to better quantitative results, and that ATE and Mask Recovery are correlated.

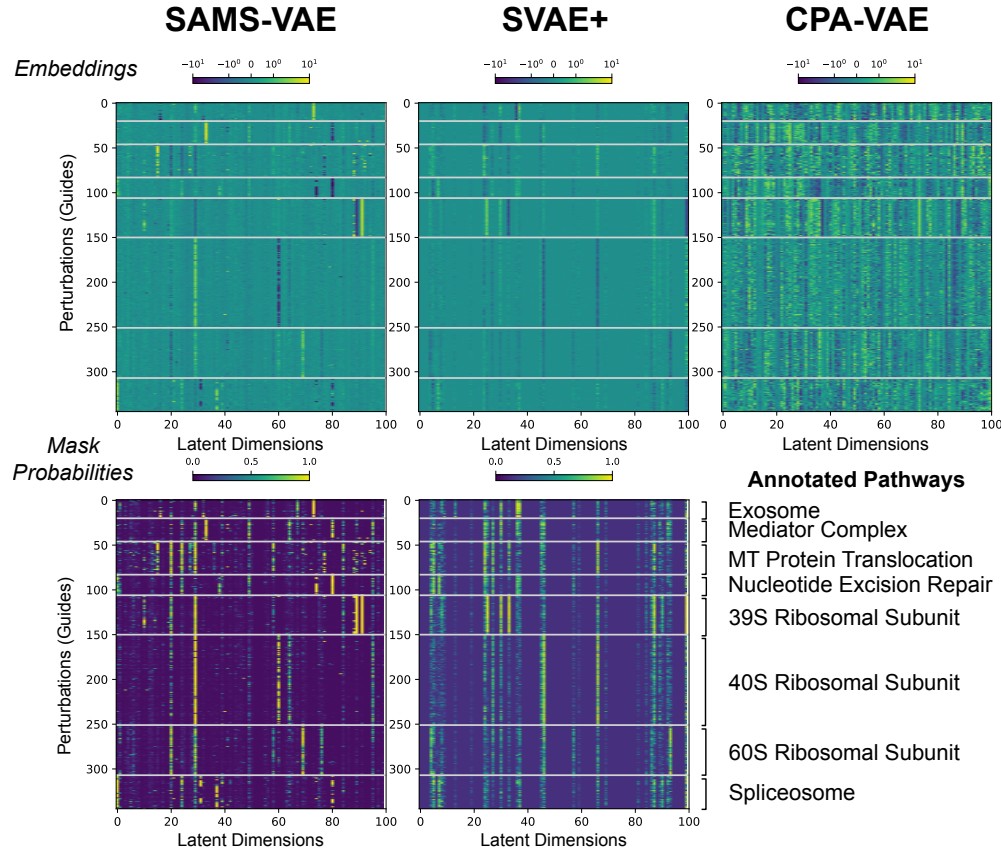

Figure 2: Visualization of inferred latent perturbation masks and embedding means for the best performing checkpoint of each model in `replogle-filtered`. We visualize the latent variables for the 345 perturbations with pathway annotations from Replogle et al. [17] and group by pathway. The SAMS-VAE and CPA-VAE models were trained with our proposed correlated inference strategy.

**Evaluation protocol** Each model is trained with a 100 dimensional latent space and MLP encoders and decoders with a single hidden layer of dimension 400 (see Section A.4 for full training details). Based on validation performance and sparsity, a $\text{Beta}(1, 2)$ prior was selected for the SVAE+ mask, and a $\text{Bern}(0.001)$ prior was selected for SAMS-VAE. For each model type, we compute the test set importance weighted ELBO as described in Section 3.1 and report the mean and standard deviation across five training runs with different random seeds. We additionally estimate the model average treatment effect with $K = 10,000$ particles as defined in Section 3.2 for the best model of each type and report the correlation between this quantity and the estimated differential expression from data.

**Results** Quantitative results are presented in Table 1. Comparing first between model types, we observe that SAMS-VAE with fully correlated inference achieves the best test IWELBO and average treatment effect correlation. Interestingly, CPA-VAE with correlated inference achieves strong test IWELBO performance but falls behind on average treatment effect prediction, while conditional VAE has weak IWELBO performance but achieves strong average treatment effect prediction. SVAE+ does not perform well on either metric in this setting.

In addition to comparing model types, we perform an ablation of SAMS-VAE and CPA-VAE inference strategies. We find that the correlated $z_{basal}$ strategy yields substantial improvements in performance for both SAMS-VAE and CPA-VAE, while the correlated $E$ strategy improvements are minor.

### 4.1.1 Recovery of biological mechanisms based on disentangled factors

**Evaluation protocol** We assess the degree to which the pattern of perturbation effects on inferred latent factors in the SAMS-VAE and SVAE+ models from the previous section are predictive of

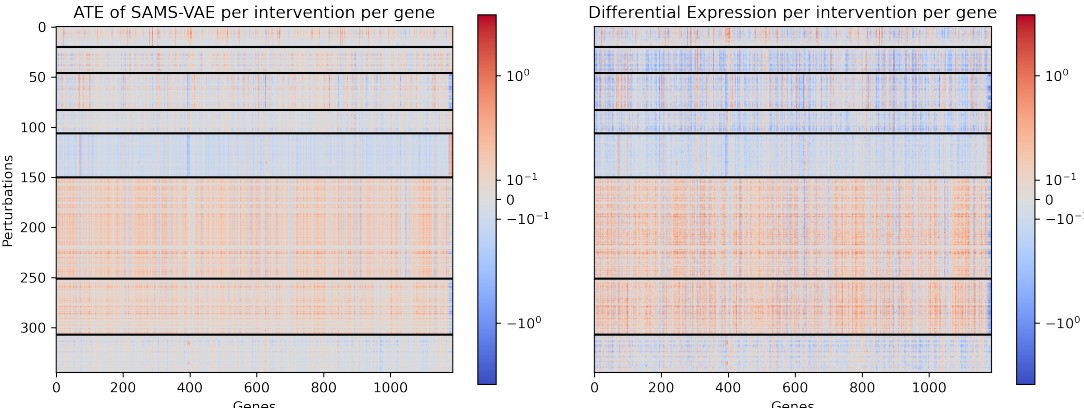

Figure 3: We visualize model-estimated treatment effects ($\text{ATE}_{\text{SAMS-VAE}}$) and data-estimated differential expression ($\text{DE}_{\text{Data}}$) for intervention-gene pairs in the Replogle experiment. We observe broad correlation (Pearson $r = 0.765$): for example, perturbations of ribosomal subunits influence on all expression broadly with matching directionality, while other guides exhibit more targeted effects.

known biological pathways as annotated by Replogle et al. [17] (345 of the 722 targeted genetic perturbations are annotated). To do so, we define an inferred binary mask of perturbation effects on latent factors by thresholding the inferred latent mask probabilities in each model at $p = 0.5$. For each model, we fit a random forest model using scikit-learn [16] to predict pathway annotations from a subset of the perturbation latent masks and assess pathway prediction accuracy on the remaining perturbations. This evaluation is performed on the best checkpoint for each model type with 10 random splits of perturbations (70% train, 30% test), and the mean and standard deviation of the pathway prediction accuracy is reported.

We also provide a set of visualizations to qualitatively assess the latent structures learned by each model. We plot the inferred masks and embeddings for SAMS-VAE, SVAE+, and CPA-VAE in Figure 2, and visualize the SAMS-VAE estimated average treatment effects and estimated differential expression corresponding to these perturbation effects in Figure 3. Hierarchical clustering and UMAP projection of the inferred perturbation embeddings are presented in Figure 7 in Section A.6.

**Results**   We observe that the latent mask inferred by SAMS-VAE is more predictive of the annotated pathways in the `replogle-filtered` dataset than that inferred by SVAE+. Additionally, we find that performing correlated inference on the perturbation embeddings improved pathway prediction performance for SAMS-VAE. Qualitatively, we observe that both SAMS-VAE and SVAE+ infer sparse masks with distinct patters between annotated pathways.

## 4.2   Modeling compositional interventions in a CRISPRa perturb-seq screen

**Dataset**   We analyze the CRISPR activation (CRISPRa) perturb-seq screen from Norman et al. [15] to assess how effectively SAMS-VAE and baselines model the effect of perturbation combinations. This screen was specifically designed with perturbations that have non-additive effects in combination, making this a challenging setting for modeling combinations. We adopt the preprocessing from [6], which contains 105 unique targeting guides applied both on their own and in 131 combinations. In total, the dataset contains 111,255 cells, each with 5,000 gene expression features.

**Evaluation protocol**   We define two tasks using this data. The first, `norman-ood`, assesses the ability of each model to predict gene expression profiles for held-out cells that have received perturbation combinations that are not included in the training set. Each model is trained on cells that received a single guide, along with [0, 25, 50, 75, 100]% of combinations. Held-out cells receiving the final 25% of combinations are used to evaluate each model. We perform this analysis for 5 random splits of the combinations. The second task, `norman-data-efficiency`, assesses how efficiently the models can learn combination phenotypes when trained on cells that have received a single guide and increasing numbers of cells sampled uniformly across all combinations. Each model is evaluated

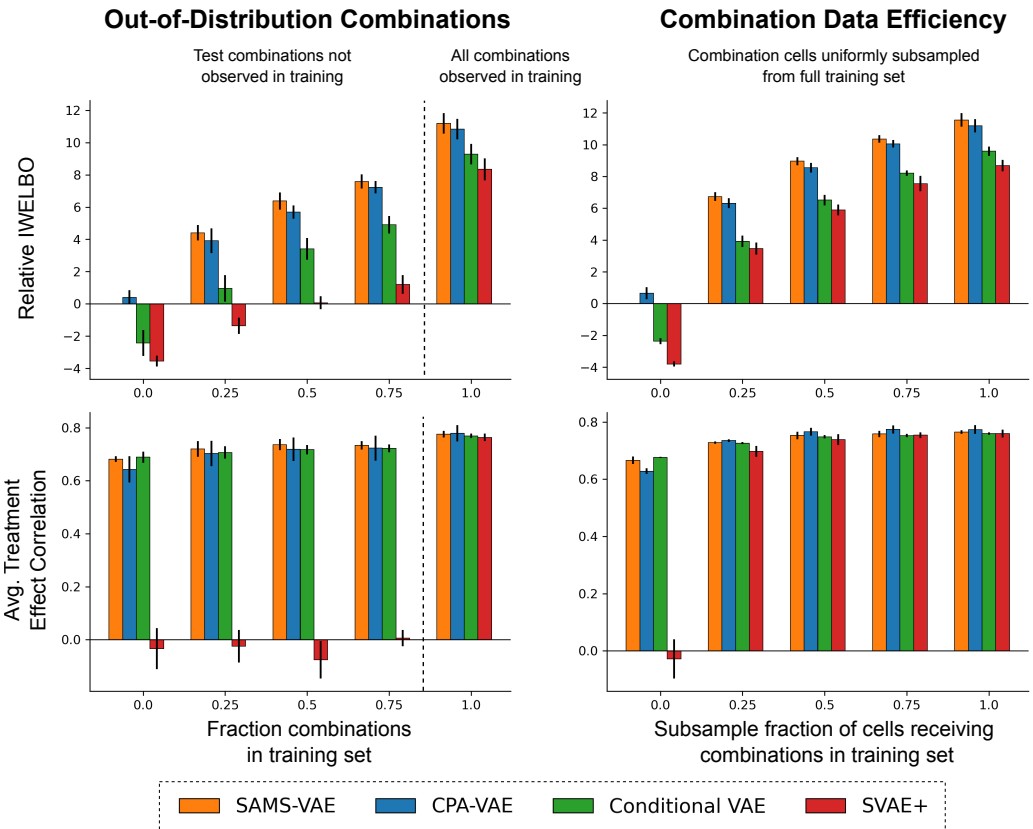

Figure 4: Results from `norman-ood` and `norman-data-efficiency` experiments. Within splits, test IWELBO values are plotted relative to the test IWELBO for SAMS-VAE trained with 0 combinations on that split (relative IWELBO) to enable comparison across splits. SAMS-VAE and CPA-VAE models are trained with the correlated inference schemes described in methods.

based on the IWELBO and ATE-Pearson on the held out test set. To compare model performance across different data splits, within each split we analyze the test IWELBO of each model relative to the test IWELBO of SAMS-VAE trained with no combinations on that split (relative IWELBO). Average treatment effects are predicted with 2,500 particles, and IWELBO values with 100 particles.

We train each model with latent dimension 200 and single hidden layer MLP encoders and decoders for 30,000 training steps. Based on validation performance, SVAE+ is trained with a $\text{Beta}(1,2)$ prior and SAMS-VAE is trained with a $\text{Bern}(0.01)$ prior.

**Results** Quantitative results are presented in Figure 4, with additional inference strategy ablations in Figure 10 in Section A.6. SAMS-VAE and CPA-VAE both achieve strong performance on the `norman-ood` task across metrics, often within 1 standard deviation of one another. Conditional VAE achieves similarly strong performance for average treatment effect prediction, though is weaker on the IWELBO metric. Unsurprisingly, SVAE+, which models combinations as totally new treatments, is unable to predict the effect of a new combination without observing it in training. We do observe that the SVAE+ likelihood still improves a small amount as more combinations are included in training set, which may be attributable to improvments in the encoder and decoder (which are shared across perturbations). These results support the utility of the compositional mechanisms in SAMS-VAE and CPA-VAE (and for encoding combinations as defined in $\boldsymbol{d}_i$ for conditional VAE).

In `norman-data-efficiency`, we observe similar trends. SAMS-VAE, CPA-VAE, and conditional VAE, which can share information across individual and combined perturbations, all achieve better ATE prediction for held out cells when less than 50% of the available combination cells. However, SVAE+ achieves similar ATE prediction correlations on this dataset when presented with sufficient

combination samples in training. Looking at the relative IWELBO values, we observe that SAMS-VAE and CPA-VAE again perform the best, with SVAE+. These results further support the utility of the additive composition mechanism from SAMS-VAE and CPA-VAE in low data settings.

## 5 Related Work

**Disentangled VAEs**   Disentangled variational auto-encoders have been proposed as early as in [7], where weak supervision was utilized to learn sparse masks over different subspaces. A popular framework for unsupervised disentanglement was proposed in [3] through a reweighting of the regularizer in the objective, but ignores weak supervision about conditions. A more comprehensive treatise and theoretical analysis of disentanglement was presented in [10]. Finally, the formal link to sparse mechanism shift and explicit causal disentanglement was also established recently in [9]. Our work shares assumptions with some of these works, in that we assume dosage is known leading to specific shifted effects per perturbation that can be used to learn disentangled factors.

**Models of Cellular Perturbation**   A popular generative modeling framework for cellular sequencing data utilizing VAEs has been proposed in [11], a model which inspired our use of their likelihood. Closer to our application on perturbed datasets is the CPA [13]. Similar to this model, we adopt the idea to disentangle cellular latent spaces into basal and perturbation latent variables. However, we pose the resulting model as a joint generative model with a rigorous inference framework and, crucially, a sparsity mechanism to disentangle the perturbation effects into subspaces related to the affected mechanisms. The recent SVAE+ [12] model is an exciting variant of [9] that utilizes disentanglement in a fashion that matches our goals. Our work differs by factorizing variation into basal and perturbation variables, adding a mechanism to compose perturbations, and in terms of inference strategies (SVAE+ learns perturbation representations by optimizing a prior). GEARS [18] leverages prior information of perturbation and gene features to predict the effect of applying new perturbations to unperturbed cells. This work instead focuses on specifying a generative model for perturbation effects with minimal assumptions, though strategies for integrating prior information on perturbations and features in specific use cases is an exciting future direction.

## 6 Conclusion

Performing unbiased scientific discovery is an aspirational goal in the field of drug discovery to detect mechanisms of action with intervenable potential. We propose a model that attempts to use few explicit assumptions about the nature of the observed data and relies heavily on a sparsity assumption and decomposition into explicit treatment effects in latent space to learn models of perturbational screening data, making it general enough for application to arbitrary data modalities and perturbation types. In this work, we apply SAMS-VAE to genetic perturbations and single-cell sequencing readouts and observe two key outcomes: improved predictive performance compared to omitting the sparsity assumption, and improved ability to recover factors correlated with real mechanisms in biological data. Our technical contributions cover both a specification of a novel sparse generative model, SAMS-VAE, as well as a suite of inference strategies for improved model fit which also apply to our baseline CPA-VAE. We also propose an evaluation strategy for perturbation models related to posterior predictive checks utilizing average treatment effects and differential expression as test statistics to perform model criticism, and observe our model performing competitively in this metric.

Such models may ultimately be useful to perform experiments in an iterative fashion, and help specify actionable hypothesis spaces for more targeted experiments down the line. Our work falls into a long line of literature on disentanglement and more recently the Sparse Mechanism Shift hypothesis related to causality, and we believe that the specific setup of SAMS-VAE will be useful in practical scenarios while being quantitatively performant across relevant tasks.

The deliberately generic assumptions we make about perturbations pose opportunities for future inquiry into more detailed aspects of such models. In specific cases, we may have prior knowledge about the nature of perturbations and their effects on the system we observe. An interesting future direction is posed in studying how perturbations may interact and compose in more complex fashion, and incorporating different forms of prior knowledge into such systems, while maintaining the ability of the system to discover knowledge and factors of variations that can be used downstream.

## Acknowledgements

We acknowledge and thank insitro for funding this work.

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
