

**Algorithm 2** CPA-VAE generative process

**Require:** $\boldsymbol{X} \in \mathbb{R}^{N \times D_x}, \boldsymbol{D} \in \{0,1\}^{N \times T}$
  **for** $t$ from 1 to $T$ **do**
    $\boldsymbol{e}_t \sim \mathcal{N}(0, I)$
  **end for**
  **for** $i$ from 1 to $N$ **do**
    $\boldsymbol{z}_i^b \sim \mathcal{N}(0, I)$
    $\boldsymbol{z}_i^p = \sum_{t=1}^{T} d_{i,t} \cdot \boldsymbol{e}_t$
    $\boldsymbol{z}_i = \boldsymbol{z}_i^b + \boldsymbol{z}_i^p$
    $\boldsymbol{x}_i \sim p(\boldsymbol{x}_i | \boldsymbol{z}_i; \boldsymbol{\theta})$
  **end for**

Figure 5: CPA-VAE represented as an generative process (left) and as a graphical model (right).

# A  Appendix

## A.1  Mini-batch optimization

In this section, we provide a detailed description of how the ELBO is computed from mini-batches for optimization.

Replacing the expressions for the generative distribution 2 and correlated variational distribution 5 in the ELBO 3, we have the following expression for the ELBO:

$$\text{ELBO}(\boldsymbol{\phi}, \boldsymbol{\theta}) = \mathbb{E}_{\boldsymbol{Z}^b, \boldsymbol{E}, \boldsymbol{M} \sim q(\cdot | \boldsymbol{X}, \boldsymbol{D}; \boldsymbol{\phi})} \log \frac{\left[\prod_{t=1}^{T} p(\boldsymbol{e}_t) p(\boldsymbol{m}_t)\right] \left[\prod_{i=1}^{N} p(\boldsymbol{z}_i^b) p(\boldsymbol{x}_i | \boldsymbol{z}_i^b, \boldsymbol{d}_i, \boldsymbol{M}, \boldsymbol{E}; \boldsymbol{\theta})\right]}{\left[\prod_{t=1}^{T} q(\boldsymbol{m}_t; \boldsymbol{\phi}) q(\boldsymbol{e}_t | \boldsymbol{m}_t; \boldsymbol{\phi})\right] \left[\prod_{i=1}^{N} q(\boldsymbol{z}_i^b | \boldsymbol{x}_i, \boldsymbol{d}_i, \boldsymbol{E}, \boldsymbol{M}; \boldsymbol{\phi})\right]}$$

During training, we iterate through shuffled versions of the training dataset and receive batches of indices $B = \{i_1, ..., i_{|B|}\}$. Let $n_t = \sum_{i=1}^{N} D_{i,t}$ be the total number of samples in the training set that have received perturbation $t$ and let $\tilde{n}_t = \sum_{i \in B} \tilde{D}_{i,t}$ be the total number of samples in the batch that have received perturbation $t$. Let $P$ be a hyperparameter *number of particles*. We compute the mini-batch loss as follows:

$$l_{mb} = \frac{1}{P} \sum_{p=1}^{P} \left[\sum_{t=1}^{T} \frac{\tilde{n}_t}{n_t} \log \frac{p(\boldsymbol{e}_t) p(\boldsymbol{m}_t)}{q(\boldsymbol{m}_t; \boldsymbol{\phi}) q(\boldsymbol{e}_t | \boldsymbol{m}_t; \boldsymbol{\phi})}\right] \left[\sum_{i \in B} \log \frac{p(\boldsymbol{x}_i | \boldsymbol{z}_i^{b^{(p)}}, \boldsymbol{d}_i, \boldsymbol{M}^{(p)}, \boldsymbol{E}^{(p)}; \boldsymbol{\theta})}{q(\boldsymbol{z}_i^{b^{(p)}} | \boldsymbol{x}_i, \boldsymbol{d}_i, \boldsymbol{M}^{(p)}, \boldsymbol{E}^{(p)}; \boldsymbol{\phi})}\right]$$

where $(\boldsymbol{Z}^{b^{(1)}}, \boldsymbol{E}^{(1)}, \boldsymbol{M}^{(1)}), ... (\boldsymbol{Z}^{b^{(P)}}, \boldsymbol{E}^{(P)}, \boldsymbol{M}^{(P)})$ are samples from the variational distribution $q(\boldsymbol{Z}^p, \boldsymbol{E}, \boldsymbol{M} | \boldsymbol{X}, \boldsymbol{D}; \boldsymbol{\phi})$. Note the reweighting term $\frac{\tilde{n}_t}{n_t}$, which maintains the ratio from the full ELBO between the prior terms on the perturbation masks and embeddings and the likelihood terms on samples that received those perturbations while ensuring that the prior terms of treatments that are not included in the mini-batch do not contribute to the mini-batch loss. Thus

$$\mathbb{E}_{B, \boldsymbol{Z}^b, \boldsymbol{E}, \boldsymbol{M}} l_{mb} = \frac{|B|}{N} ELBO(\boldsymbol{\phi}, \boldsymbol{\theta}; X, D).$$

## A.2  CPA-VAE

Variational Family for CPA-VAE:

$$q(\boldsymbol{Z}^b, \boldsymbol{E} | \boldsymbol{X}, \boldsymbol{D}; \boldsymbol{\phi}) = \left[\prod_{t=1}^{T} q(\boldsymbol{e}_t; \boldsymbol{\phi})\right] \left[\prod_{i=1}^{N} q(\boldsymbol{z}_i^b | \boldsymbol{x}_i, \boldsymbol{e}_t; \boldsymbol{\phi})\right]. \tag{8}$$

### A.3 Detailed comparison with prior work

#### A.3.1 Compositional Perturbation Autoencoder

Lotfollahi et al. [12] present Compositional Perturbation Autoencoder (CPA), a method for modeling perturbation effects in cellular data. CPA has many similarities to SAMS-VAE and was a key inspiration: CPA learns to encode observed cells into basal vectors, which are then added to learned embeddings of perturbations and covariates to define a cell latent state, which is mapped to a predicted phenotype through a neural network decoder. The model can be used to predict the effect of new treatments by encoding observed cells to their latent basal states, shifting by the corresponding latent embedding, and decoding.

However, there are a few key differences between our methods and CPA. First, our method explicitly models sparsity in latent perturbation effects. Second, SAMS-VAE (and CPA-VAE) are fully defined generative models. CPA does not specify a prior for the latent basal state, so any predictions must start from an observed cell which is encoded; by contrast, SAMS-VAE specifies prior probability distributions for all variables. This has a few benefits, including 1) allowing samples of $p(x|d)$ without having to define reference cells (we can sample from our prior on latent basal states, while CPA needs to encode other observed cells to generate samples) and 2) allowing for estimates of the likelihood $p(x|d)$. Third, CPA requires an adversarial network to try to learn latent variables that are not correlated with perturbations (we apply variational inference to our generative model, which encourages this property). Fourth, CPA has a mechanism to encode dosages nonlinearly in their latent space–this could be a useful modular addition to our work but was not needed for the datasets we consider, which have have binary dosages (present or absent).

#### A.3.2 SVAE+

SAMS-VAE also shares similarities with SVAE+ [11]. SVAE+ is a generative model for modeling perturbation effects in cells that explicitly models sparsity with a mask and embedding mechanism. However, there are a couple key differences. SVAE+ does not have a mechanism to compose interventions, whereas SAMS-VAE models a latent space where perturbations compose additively. Second, SVAE+ does not explicitly model a cell's latent basal state as a random variable: each cell has its full latent embedding sampled from a learned prior (using type II maximum likelihood) that is conditioned on the received treatment. This learned prior, along with the variational inference families considered, are also substantial differences from SAMS-VAE.

#### A.3.3 Summary

The key contribution of SAMS-VAE is a generative model which combines useful key principles that have been applied in prior work: like CPA, it models a cell's latent state as a sum of a cell basal state and learned perturbation embeddings, and like SVAE+ it is a generative model that explicitly models sparsity in perturbation embeddings with a masking mechanism. We additionally gain some performance improvements through careful inference using standard ML techniques.

### A.4 Additional experiment details

**Perturbseq data normalization** We train all generative models directly on transcript counts as described in Section 2.2. Input transcript counts are log transformed and standardized when provided to the generative model encoders. to When computing differential expression and average treatment effects, expression values are normalized by library size for each sample.

**Encoder and decoder architectures** All model encoders and decoders are fully connect neural networks with residual connections and leaky ReLU non-linearities.

**Conditional VAE treatment representation** In the conditional VAE model, treatment dosage vector $d_i$ is directly concatenated to the latent state $z_i$ for decoding. As described in methods, $d_{i,j} = 1$ if sample $i$ received perturbation $j$ and is 0 otherwise.

**Replogle experiment additional details** Each model was optimized with the Adam optimizer for 150,000 steps with batch size 512, learning rate 0.0003, and weight decay 1E-6. Checkpoints were saved every 2,000 training steps, and the checkpoint with the best validation ELBO was used in test evaluation. Following the original paper, priors of the form $\text{Beta}(1, K)$ were considered for SVAE+. Specifically we consider $K \in \{2, 5, 10\}$ and find $K = 2$ performs best. We consider priors of $\text{Bern}(\alpha)$ for $\alpha \in \{0.1, 0.01, 0.001\}$ for SAMS-VAE. We find that the validation IWELBO is very similar across this range, and choose $\alpha = 0.001$ based on our objective of identifying sparse latent masks.

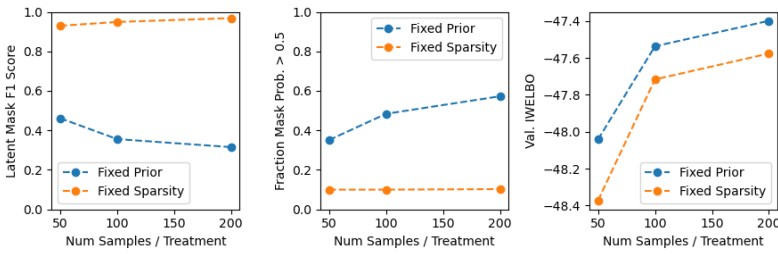

Figure 6: Results of experiment with SAMS-VAE using simulation data

## A.5 Simulation analysis

We perform a brief analysis of SAMS-VAE using simulated data to assess the relationship between sparsity hyperparameters and mask recovery. We apply the simulation framework introduced in Lopez et al. [11] with minor modifications. Briefly, we simulate data from the SAMS-VAE generative process with latent dimension 15, latent perturbation masks sampled from $Bern(0.1)$, and latent embeddings sampled from $N(5, 0.5)$ (large latent offsets matching prior work). Latent basal variation is sampled from $N(0, 1)$. Observations (50 features per cell) are sampled from a Gaussian likelihood $N(\mu_\theta(z_i), \sigma^2 I)$, where $\mu_\theta$ is an MLP with 2 20-dimensional hidden layers. Following Lopez et al. [11], the weights of the decoder MLP are initialized to orthogonal matrices for injectivity. $\sigma^2$ is set such that 80% of the variance in each feature is due to $\mu_\theta(z)$. We emphasize that we do not expect this simulation to correspond to the true generative process of biological datasets and focus on using the simulation to explore how prior hyperparameters relate to inferred masks.

We generate simulated training datasets with 50, 100, and 200 samples per treatment. We then fit SAMS-VAE in two settings: **fixed prior**, where the mask Bernoulli prior is set to $\alpha = 0.1$ and **fixed sparsity**, where the mask prior probability is adjusted so that the inferred mask has sparsity close to 0.1. We set the prior on the perturbation embeddings as a relatively uninformative prior $N(0, 10)$ to accommodate the large offsets, and set the encoder and decoder neural networks to two-layer MLPs with 100 hidden dimensions. Following Lopez et al. [11], we compute the F1 score between the inferred and simulated masks after thresholding the inferred mask at $p = 0.5$ and permuting columns to maximize the true positive rate between the true and inferred binary masks.

Focusing first on the fixed prior experiment, we observe that the inferred mask is more dense than the simulated mask, and that the mask becomes more dense as the sample size increases. Looking at the equation for the SAMS-VAE ELBO (and IWELBO), we observe that the the prior terms for global variables do not scale with sample size, while the observation likelihoods do. Thus, the loss increasingly prioritizes observation likelihoods over the global variable priors. This is generally a desirable property: for example, we want the priors on perturbation embeddings to be weighed against and updated based on all of the samples we observe. However, given the challenges of model mismatch, stochastic optimization, and approximate inference, this can lead to dense masks, revealing a limitation to the semantics of framing mask sparsity as a global prior when we aim to assert sparsity for the purposes of downstream analysis. Motivated by this observation, we also consider adjusting the prior strength or reweighting the ELBO to achieve a desired level of sparsity. In this experimental setup, we find that setting the mask prior probability to $\alpha = 10^{-\frac{9}{50}n_t}$, where $n_t$ is the number of samples for each treatment, maintains a mask sparsity of approximately 0.1 and effectively recovers the simulated mask (F1 > 0.9). These conclusions align with challenges identified by Lopez et al. [11].

## A.6 Supplementary Figures

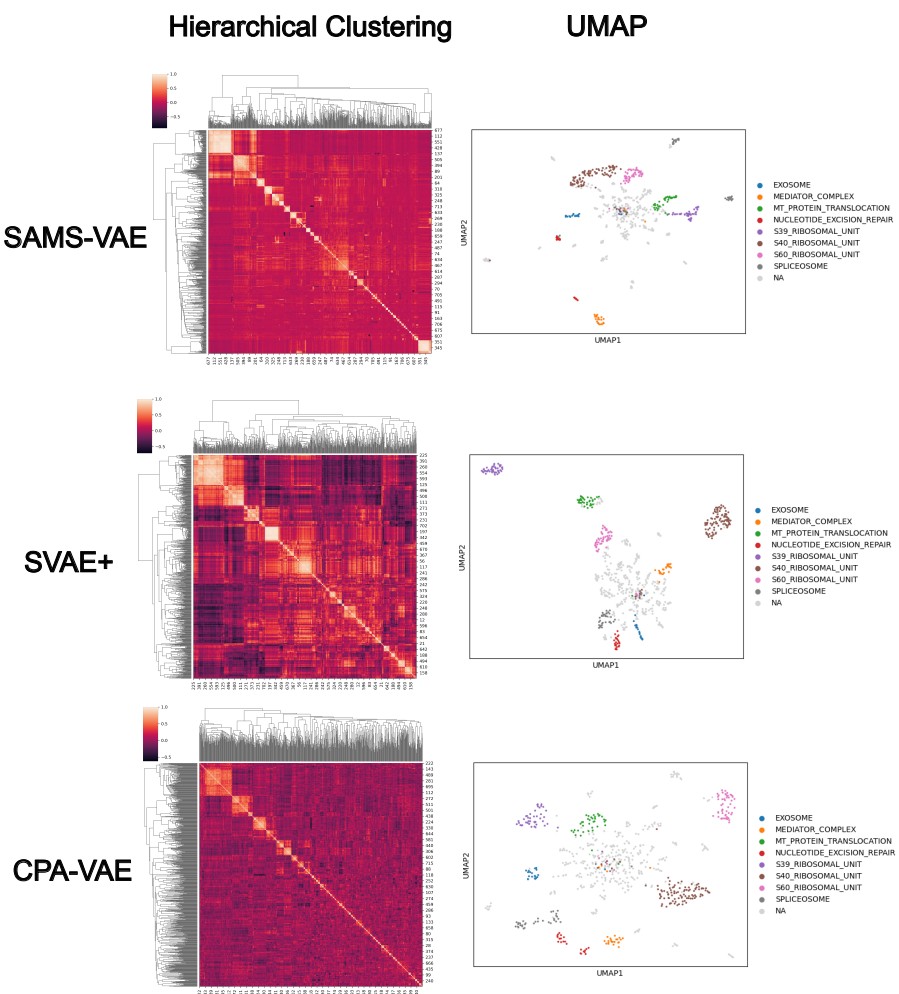

Figure 7: Hierarchical clustering and UMAP of inferred latent embeddings in the `replogle-filtered` dataset. SAMS-VAE and CPA-VAE models were trained with the fully correlated inference strategy. Pathway annotations are provided in Replogle et al. [16].
.

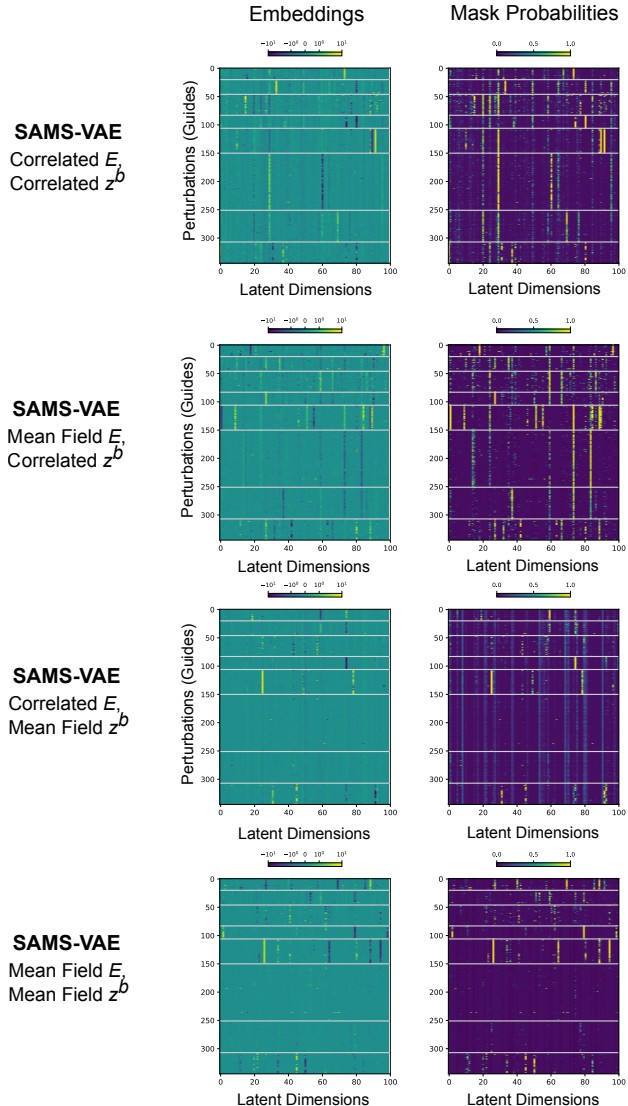

Figure 8: Hierarchical clustering and UMAP of inferred latent embeddings in the `replogle-filtered` dataset. SAMS-VAE and CPA-VAE models were trained with the fully correlated inference strategy. Pathway annotations are provided in Replogle et al. [16].

.

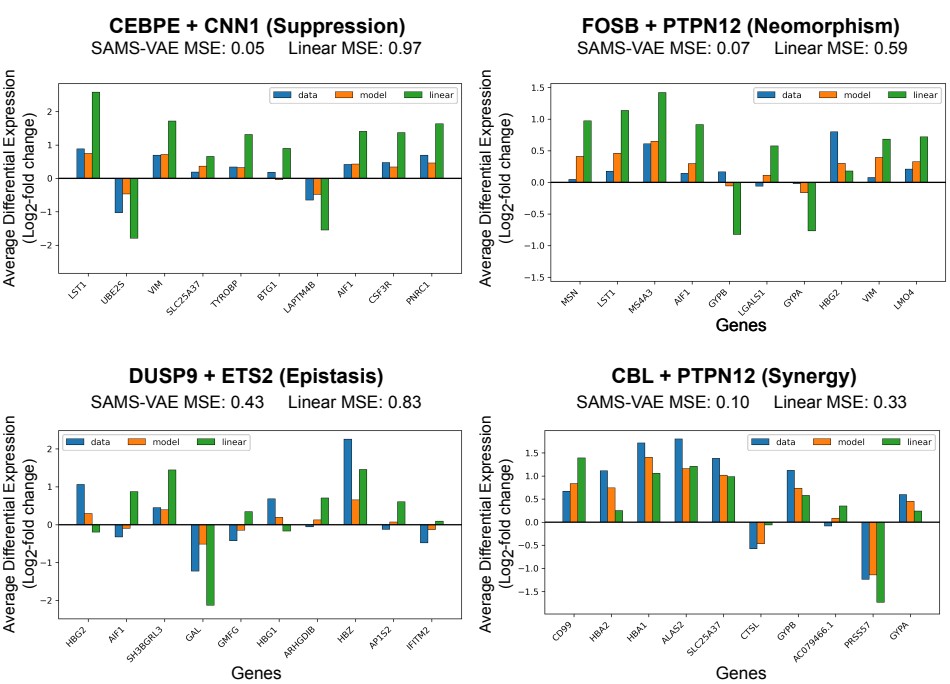

Figure 9: Visualization of SAMS-VAE predictions of held-out combinations with strong nonlinear effects in `norman-ood`. We visualize perturbation combinations and corresponding gene expression features identified to exhibit strong nonlinear genetic interactions in Roohani et al. [17]. We observe that SAMS-VAE improves prediction beyond a naive linear model that predicts the sum of each perturbation independently, though it still faces difficult predicting some nonlinear interactions (e.g DUSP9 + ETS2)

.

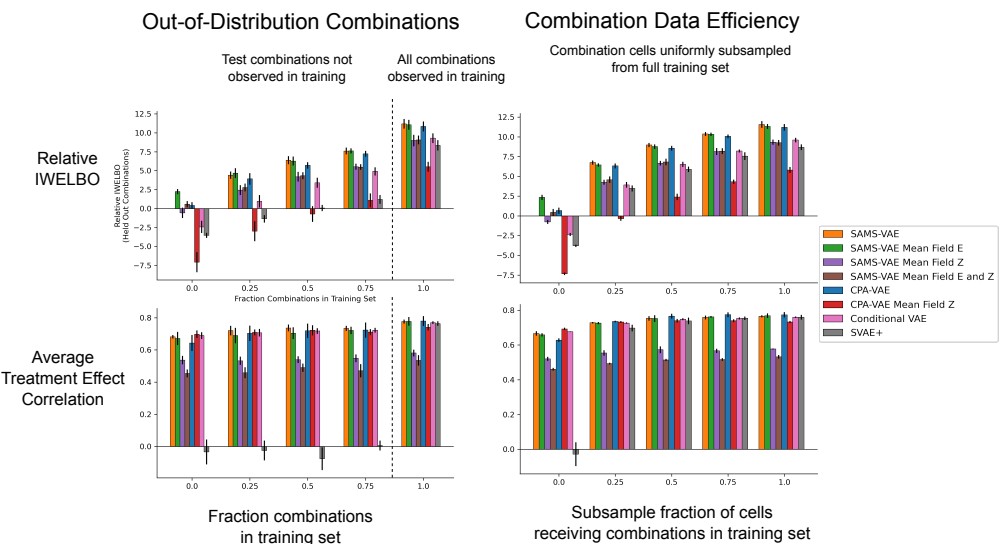

Figure 10: Extended ablation results from `norman-ood` and `norman-data-efficiency` experiments (see 4). Within splits, test IWELBO values are plotted relative to the test IWELBO for SAMS-VAE trained with 0 combinations on that split (relative IWELBO) to enable comparison across splits.

.