# OpenReview forum: "Modelling Cellular Perturbations with the Sparse Additive Mechanism Shift Variational Autoencoder"
_NeurIPS.cc/2023/Conference — NeurIPS 2023 poster_

### Official Review · Reviewer_TMYT · 2023-07-02

**Soundness:** 2 fair
**Presentation:** 3 good
**Contribution:** 2 fair
**Rating:** 4
**Confidence:** 3

**Summary:**

This paper presents a generative model for single cell RNA sequence data that models perturbations as sparse linear offsets in latent space. They derive a variational inference algorithm for their model and evaluate it on a couple of perturb-seq datasets where the perturbations are CRISPR knockouts of genes. They find that the inferred latent mask can be used for predicting known biological pathways, and they get some improvement in out-of-distribution prediction of gene expression.

**Strengths:**

* The paper does a good job of articulating appropriate assumptions for expression data.
* The empirical results look good relatively to a variant on Lotfollahi et al.'s CPA and Lopez et al.'s sVAE+ in the sense that they show that you can enforce sparsity while maintaining or slightly improving likelihood (or at least an importance weighted bound on likelihood).

**Weaknesses:**

This paper is tricky to evaluate because it leans heavily on the sparsity story (because otherwise it is a very small modification of Lotfollahi et al.'s CPA), so they motivate the work by connecting it to ideas from causal representation learning (sparse mechanism shift hypothesis) where the goal is to recover latent variables. But I think that's a risky strategy: making a model sparse does not automatically make it causal - especially when there's no reason to believe that the sparsity corresponds to how you would expect latent variables to behave. Given cascading nonlinear feedback of cell biology, I don't think anyone seriously thinks that the underlying biological mechanisms are well described by sparse linear offsets. But, from the empirical results, it does seem like sparsity is a useful way of constraining the model's capacity, and it provides the added benefit of making the model more interpretable.

I think to properly evaluate the utility of sparsity, I would want to see how a biologist could use the output of this model to discover a pathway. I know that's what the annotated pathway experiments were meant to illustrate, but they used labeled data to train the random forest, and when discovering a novel pathway, you wouldn't have these labels. The plots in figure 2 suggest that this may be possible (there's a lot of shared structure within pathways), but I'm not sure how much of that structure is a function of the fact that the probabilities are order by pathway. Unfortunately, I think doing this kind of an evaluation really well would probably turn this into a paper better suited to a bio venue because they could better evaluate how realistic the proposed procedure is.

**Questions:**

* What do the clusters in the hierarchical clustering correspond to? Do they have a biological interpretation? And if so, could this method be used for discovery via clustering?

**Limitations:**

Given the causal framing of the introduction, I would have liked to see some more discussion about what biological processes / features the authors believe the latent variables should correspond to, and what simplifying assumptions their model makes with reference to these beliefs.

---

> ### Author Rebuttal · Authors · 2023-08-10
>
> We thank the reviewer for their deep feedback and thoughtful response about the core of our proposed model. We believe we can address your concerns resoundingly and hope to convince you to raise your score. Please read our general response since it addresses your main concern as well.
>
> **Results vs. CPA and SVAE+:**
> We thank the reviewer for pointing this out as a positive. However, as we discussed with R3, we believe our results are not just slightly better, but significantly so. CPA-VAE only is close to SAMS-VAE when we apply correlated inference strategies to that model, which is also one of our contributions from Sec 2.3. The baseline version of the model using mean-field amortized inference, which is a fair statement is the baseline, performs significantly worse as we show in Table 1. In Fig 4 SAMS-VAE also is the best performer, although it is compared to CPA-VAE with correlated inference we propose in this paper. We did not report mean field there to ablate the effect of the sparsity itself.
> We put a lot of work into getting the interactions in the posterior between the basal state and the treatment embedding right, and later adding the mask to that and maintaining that performance. Out of the box these objects are worse behaved, in performance and interpretability as also the pathway prediction task in in Table 1 shows, and as such we would like to highlight our work on approximate inference in Sec 2.3 by proposing a suite of scalable improvements to the variational family for this model as an important contribution to this model class.
> SVAE+ is the most recent published comparator here as well and underperforms compared to SAMS-VAE which also models sparsity.
> We thank you for pointing out that we managed to improve results compared to CPA-VAE and SVAE+ while incorporating the best inductive biases of both models, it absolutely benefits from the inference work.
>
> **Evaluation of Sparsity vs Causality:**
> We also discuss this with R3 and will write a general overview, but we wanted to give you a personalized response here as well.
> We believe this topic is probably the most important reason why we built this model and want to convince you of its relevance.
> First of all, we do not make claims about causality in our work. We explicitly call it disentanglement, where comparable papers including SVAE+ call their model causal, because we do not believe we can assure causality in a narrow sense. And we do not intend to.
> We fully agree that the nonlinear and obscure nature of cell biology cannot be captured easily in a mask. Some works attempt to model it by regulatory network inference, which is a hard task and reduces to (causal) structure learning.
> What we are trying to achieve in our work is to provide an unbiased way to characterize mechanisms of action as subspaces in an embedding and disentangle them from each other and from the natural variation of a cell (i.e. cell-cycle etc.) that is not perturbation specific.
> Our model manages to capture a diverse set of perturbations, and we show that the masks it learns are highly predictive of pathways in Table 1 and Sec. 3.1.1.
> However, calling the masks pathways would be inaccurate, since some treatments in this model could easily be chemical in nature, and would not strictly correspond to one pathway.
> However, as we see these “mechanism” do have interesting structure:  when observing Fig. 2, we would highlight that it becomes clear that our model separately models activity of a perturbation on an embedding dimension from the direction/effect of that perturbation.
> For instance, different ribosomal units have largely overlapping masks, interestingly shifting in some subspaces for each unit but sharing in some, while the embeddings for some shared subspaces have the opposite sign, for example guides 100-150 modeling S39 ribosomal unit have the opposite sign in treatment embedding 63, but a consistent activation of mask with other ribosomal units below them.
> This is a neurips paper, so we do not emphasize biological interpretability here and wanted to highlight the ability of the model to perform strongly and to disentangle, ideally in rigorous quantifiable ways. Using such structures for insight generation for a biology journal is of course of interest to future work, but in this manuscript we wanted to establish its predictive power and the many details in inference and modeling necessary to achieve those.
> But from a practical perspective, this model is designed to help study biology in a screen, so that specific effects can be surfaced and quantified. One can then also vary the specific subspaces belonging to a perturbation and observe what effects they have in observed space.
> Moreover, the model in its current form is applicable to any readout, be it images, transcriptomics, or whatever, given any type of perturbation, chemical or genetic or otherwise, and can point to subspaces of variation that uniquely capture the relevant bits of information.
> For instance, we are interested in analyzing the CPJump dataset ( https://jump-cellpainting.broadinstitute.org/ ) which would require an unbiased model that does not strictly just capture pathways or other pre-specified prior knowledge, and may want to study the overlap of effects between genetic and chemical perturbations, which in our model are trivial to jointly incorporate without any changes.
> Returning to our manuscript, our pathway prediction task convincingly shows that the disentanglement in our paper is a lot better at recovering previously known pathways, but in practice we agree we would not care about that. However, in order to gain confidence in a model that we use for novel discovery as hinted at above, we would clearly want to evaluate it on positive controls such as known pathways to establish its utility.

---

> > ### Comment · Reviewer_TMYT · 2023-08-16
> > **Disentanglement / causal representation learning.**
> >
> > I wanted to respond to the second section on disentanglement vs causality. I think your goals of disentangling "mechanisms of action as subspaces in an embedding" make a lot of sense and is likely to be useful. That said, the trouble with connecting with causal representation learning too strongly is that a key message of that literature is that achieving disentanglement is far more subtle than one might expect. For example, BetaVAE aims to disentangle data into independent factors of variation --- which is a reasonable goal --- but the main theorem in Locatello et al [2019] (which you cite in the intro) shows that even though BetaVAE's goals are reasonable (and we have empirical examples of it working on some datasets), this form of disentanglement is impossible from IID data without further assumptions. The main role for identifiability proof is to characterize sufficient conditions for when such reasonable sounding methods will actually succeed (in the asymptotic limit!).
> >
> > I think adding a limitations section that makes explicit that, while this approach seems to be successful at disentangling, more theoretical work is need to characterize when this will be successful would address my concern in this regard.

---

> > > ### Author Response · Authors · 2023-08-16
> > > **Thanks for your comment**
> > >
> > > Dear reviewer,
> > >
> > > Thank you for your insightful response.
> > >
> > > We certainly agree with you on the trickiness of the word "causality" in this case.
> > > However, in our work I wanted to point out that we are very different to betaVAE and more similar to old work by Karaletsos et al we cite using weak supervision to disentangle.
> > > We use an observed signal about a concrete variable (the treatment dosage d) that is being changed, while betaVAE is fully unsupervised.
> > > Much like in the works of Locatello and Karaletsos, this suggests that one key aspect of the system is changing, and we disentangle to detect it.
> > > We do not aim to provide proof of causality in this work, but believe our disentanglement work neatly fits into this literature and shares assumptions in that we test narrowly for cases where we shift explicit causal factors through interventions (i.e. treatments) and learn which latent dimensions minimally have to be perturbed to map to that.
> > >
> > > We believe your suggestion to explain this in a limitations section is very good and would happily follow, but we just wanted to clarify that we are in the setting of (weakly-) supervised disentanglement with a known intervention (that has an unknown effect we aim to characterize in latent space) and not in the setting of betaVAE.

---

### Official Review · Reviewer_Aahc · 2023-07-07

**Soundness:** 3 good
**Presentation:** 2 fair
**Contribution:** 2 fair
**Rating:** 5
**Confidence:** 3

**Summary:**

The paper presents an extension of the VAE model by incorporating sparse perturbations of the latent variables, based on the sparse mechanism shift assumption. The proposed method models the latent state of perturbed samples as a combination of a local latent variable capturing sample-specific variation and sparse global variables of latent intervention. These global variables are modeled as the pointwise product between latent perturbation variables and a binary mask. The approach is evaluated using a single cell sequencing dataset to assess its generalization capability under perturbation and its ability to recover latent variations.

**Strengths:**

- The approach exhibits a good effectiveness and outperforms baselines such as SVAE+ significantly.
- The paper is written in a overall clear manner.
- The paper presents an original combination of previous models, such as the CPA-VAE and SVAE+ models.


**Weaknesses:**

- The major weakness in my opinion is that is not entirely clear what are the benefits of the proposed model to the  CPA-VAE  baseline. To elaborate the performance differences compared to the CPA-VAE model appear to be somewhat insignificant in all experiments. It would be beneficial to clearly state the major benefits of employing the proposed model over the CPA-VAE model. Additionally, it would be useful to perform experiments that demonstrate the advantages of the proposed model over CPA-VAE.
- The experimental section requires some polishing, such as referencing Figure 2 and Figure 3 in the text. Furthermore, the "todo" annotations in the Figure 4 caption should be removed.
- Accessibility could be improved:  it would be useful to include ( e.g. in the appendix) an introduction and more in depth explanations to the biological tasks addressed in order to make the paper self contained and accessible to a non biology experts readers.

*Minor:*

 I noticed a few typos:

Line 91: "functiuon" should be "function."

Line 105: "that aim" should be "that aims."


**Questions:**

- Could the authors provide further elaboration on the rationale behind choosing the additive structure of perturbations in the latent space? Are there alternative choices that could be considered?
- What motivated the preference for the importance weighted ELBO as an evaluation metric over the standard ELBO measure?
- What are other practical problems or applications that could be addressed by the proposed model? If the model is thought mainly for solving single cell predicition problems, wouldn't it better to incorporate domain knowledge in it?


**Limitations:**

The manuscript addresses the limitations adequately.

---

> ### Author Rebuttal · Authors · 2023-08-10
>
> We thank the reviewer for their feedback and hope to convince them to improve their score:
>
> **Benefits compared to baselines:**
> We apologize for the confusion, we must have not explained the results well enough in our experimental section. CPA-VAE is a model we propose to turn CPA into a generative model, imbue it with the same likelihood and make its quantitative evaluation comparable.
> At baseline, similar to the structure of CPA, CPA-VAE uses mean-field amortized inference (standard VAE inference) and obtains a Test IWELBO of -1760.06. One of our core contributions is the inference machinery we invent in Section 2.3, and when we utilize correlated inference for CPA-VAE we can push it to -1756.72. That is based on improvements to the inference we made for SAMS-VAE, that we -for fairness- also throw into a model without sparsity so we can test the value of sparsity entirely separately, but already that performance did not previously exist in this field without us even looking into sparsity. For comparison, the most recent competitor SVAE+ (published in early 2023) obtains a score of -1759.23.
> We then note that with the improved inference SAMS-VAE utilizing the extra sparsity still outperforms both on raw predictive power with an IWELBO of -1756.11, while adding a considerable amount of structure and interpretability to CPA-VAE. Separately, we evaluated the models that perform disentanglement in the pathway prediction task and see that compared to SVAE+ at 0.69 our model reaches 0.9 accuracy, which is a dramatic improvement on a task related to semantics and interpretability for a directly comparable model class while also improving predictive accuracy on the data.
> Lastly, we evaluated our model on combinatorial tasks in Fig. 4, and show that SAMS-VAE outperforms CPA-VAE (both with the best possible inference strategy we propose, not at baseline for CPA-VAE to maximize the quality of the baseline) slightly, and SVAE+ soundly.
> We again remind the reviewer that the best version of CPA-VAE utilizes a major contribution of ours by using correlated inference, and at baseline that model is significantly worse.
> We hope this clarifies the convincing nature of our quantitative improvements, gains in interpretability, and overall superior performance of our contributions, and we will absolutely take a pass at the experimental section to make this more apparent and explain it better.
> We thank the reviewer for sharing their confusion about that, we underestimated the need to explain the separate contributions in improved approximate inference and improved modeling by introducing sparsity and hope our explanation clarifies that the combination of both is what makes this performance so strong.
> In the same spirit we thank the reviewer for the feedback on Fig. 2 and 4, and emphasize that we agree we will improve the explanations in the experimental section to make the results “pop” more as we hope we demonstrated they do in their present form and link to the Figures better and polish writing overall.
>
> **More biology background:**
> We believe this is a fantastic suggestion and will follow it, the appendix offers plenty of space for such an explanation.
> We will also share a few lines for a general rebuttal, explaining the setting of a screen in drug discovery as a core example where such a model would be directly applicable in practice, and think our readers would benefit from a tighter intuitive explanation of a few of these terms.
> While the core contribution of this work is the inference machinery and the new model utilizing sparsity, understanding the setting of our experiments with more clarity could be of further secondary utility to a broader audience that might be interested in further methodological advancements that might benefit such a setting and requires some more intuition to access it.
>
> **Additive structure in latent space:**
> In some of our our experiments, in particular in experiment 3.2 using compositions of treatments, multiple perturbations are applied to cells at the same time.
> While baseline models such as SVAE+ treat combinations as a new “one-hot treatment” with no sharing with individual treatments, we can explicitly model each treatment and its masks and embeddings separately and establish composition via addition. An underlying assumption is here that addition is a good way to model composition, which in the case of independent perturbations may be the case, and in the case of interactions between treatments less so. In our discussion we note that we leave it for future work to study richer mechanisms in more detail for modeling composability, and also note that our proposed method already works better than other existing prior work we compare to and forms a sound baseline for further inquiry into this specific problem. Interesting dependent correlations could occur when considering drug cocktails with interacting effects while trying to optimize some fitness response.
>
> **IWELBO vs ELBO:**
> The ELBO (evidence lower bound) is not a quantity that permits strict comparison between models. While one can say “model A is no worse than ELBO_A, and model B is no worse than ELBO_B”, the nature of a lower bound is that this can be arbitrarily far from the true performance of the model and gives no guarantees for relative rankings of actual performance. As such, in an effort to report approximations to the actual marginal likelihood of the data logp(y|d), we utilize importance weighting to obtain the IWELBO to report quantities which are much closer to a comparable quantity and approximate the marginal loglikelihood stochastically. This was also performed in the SVAE+ paper, and is common practice in variational inference papers.
>
> **Other problems vs domain knowledge:**
> This is an excellent question, thank you for posing it. Since R4 posed a similar question, we will refer to it in the general overview, please read our response there, and please feel free to read our response to R4.

---

> > ### Comment · Reviewer_Aahc · 2023-08-18
> > **Response to rebuttal**
> >
> > I thank the authors for answering my questions and concerns, and I apologize for the misunderstanding with the baseline CPA-VAE.  I updated my score accordingly.
> >
> > Concerning the relation with methods that incorporate domain knowledge:
> >  - I think that the paper would benefit a lot in reporting some comparisons with methods that incorporates domain knowledge in the method, or at least a in depth discussion on the expected performance of them. This is because:
> >      - It is expected that the method with domain knowledge will perform better, but the gap in performance is informative.
> >      - It is very relevant to check in which cases the data driven method outperform the domain informed one, giving evidence for the former to find patterns that allows to make good predictions, which the domain informed model fails to find.
> >      - It is important to incorporate the discussion on data driven vs domain informed anyway, highlighting cases where the former can be better and the latter could be too expensive or limited.
> >
> > In general I agree with the other reviewers and authors considerations that, while the formulation is general, the method is targeted for biological applications and this should be reflected in the paper. Without results (theoretical or experimental on the right benchmarks) on disentanglement and/or causality, not much can be said about it.
> > Having a general formulation is for sure a plus, but I think that the focus should be on biological tasks and for this reason, I think that the paper would benefit from the points raised above.

---

> > > ### Author Response · Authors · 2023-08-20
> > > **Response/Clarification to the comment**
> > >
> > > Thank you for the feedback and for updating your score, we appreciate it.
> > > Regarding your questions:
> > > + Could you please point us to papers incorporating domain knowledge for perturbation modeling we should discuss? We’d be very happy to do so, but are not aware of relevant literature that studies treatment responses in cells beyond the works we have referred to.
> > > + We agree that the discussion on data-driven vs domain-informed is important, we have alluded to this also in our last paragraph of the paper (line 272-277) discussing the value of domain knowledge to be added in the future. Could you please let us know if this goes in the direction you were thinking about?
> > > + We are happy to add a paragraph linking the scenario of our paper to prior work discussing the feasibility of causal disentanglement given weak supervision (i.e. varying individual factors of variation in the data generation such as Locatello et al discuss is our scenario given that we study cells under explicit perturbations with unknown effects). This can be a more extensive section on limitations and assumptions to clarify this further.
> > >
> > > Again, thank you for your valuable feedback.

---

### Official Review · Reviewer_y8vh · 2023-07-09

**Soundness:** 2 fair
**Presentation:** 3 good
**Contribution:** 2 fair
**Rating:** 6
**Confidence:** 2

**Summary:**

This paper proposes a new VAE-based model for learning disentangled representation, dubbed Sparse Additive Mechanism Shift Variational Autoencoder (SAMS-VAE). The model is specifically designed for datasets under perturbation; the dataset has the form of $(x_i,d_i)$ where $d_i \in \{0,1\}^T$ indicates the perturbation for sample $i$. Though the generative model can be general, here the authors design it for a specific application which is scRNA-sequential data. The model consistent of a set of global variables that correspond to the perturbations as well as their sparsity as well as a set of local variables correspond to noise for each sample. The authors then perform a set of experiments on the genome-wide CRISPR interference (CRISPRi) perturb-seq dataset from Replogle et al and the one from Norman et al. The authors then evaluate their model based on loglikelihood, interpretability of the latent space (based on productiveness of annotated biological pathways), and generalization to unseen data

**Strengths:**

- The paper was easy to read for the most part. The training details as well the model design is relatively clear. I applaud the authors for making the paper as reproducible as possible.

- Though scRNA-sequential data is not my area, I feel that it is an understudied subject for generative models that could benefit from more research and it is something that will be of interest to the NeurIPS community.

- To the best of my knowledge , the paper contains no technical flaws.

**Weaknesses:**

- While the language of the paper is clear, I don't think the paper is currently self-contained particularly with regards to dismantlement. There are many definitions and notions of disentanglement out there (e.g. independence, equivariance, permutation, linear disentanglement, ...). If you are basing yours on sparse mechanism shift, then it would be helpful to provide some background where you clearly define these concepts rather than immediately jumping into defining your model.

- Relating to the previous point, I missed the motivation for some of the design choices. For example, why use addition to combine the latent? Should the sparsity variable $m_t$ be independent for each dimension?

- Some parts of the results is not very clear for me to draw conclusions from. Neither Figure 2 or 3 are discussed in the main paper. What are the variables in Figure 2? Is the top $e_t$? Why the text says there are 722 perturbation but the figure only shows around 350? At the bottom, the mask is applied to what exactly? Also based on this figure, I see no qualitative difference between SAMS-VAE and SVAE+. As for figure 3, you should also show umap of raw the data and the baselines for comparisons.

- The title for this paper is currently abit too general. This a very specific model that targets perturbation datasets and is not a general objective that can be applied to any VAE. I would consider changing the title.

- Please make sure that IWELBO equation that is not outside the margins on page 4

**Questions:**

See Weaknesses

**Limitations:**

To the best of my knowledge, this paper poses no potential negative societal impact.

---

> ### Author Rebuttal · Authors · 2023-08-10
>
> We thank the reviewer for their positive review and their detailed suggestions. We will address their feedback in the following:
>
>
> **Disentanglement/Self-Contained:**
> While we believe that we cite a rich set of papers explaining the setting we are based on, in particular the recent SVAE+ by Lopez et al which also models disentanglement in cells and was published in 2023, is inspired by Sparse Mechanism Shift, but underperforms our work, we are happy to take the feedback and will include more explanation over our assumptions and the setting in the paper. For now, we explicitly do not make strong causal statements or limiting assumptions of the nature of linear disentanglement etc., We propose a VAE-style model with a specific, novel structure and stick to a subspace inference task with a binary mask over the latent space per treatment (where each dimension is independent in the prior, but some of our inference methods correlate the dimensions in the posterior).
> We model the existence of multiple treatment as additive perturbations, allowing superpositions of disentangled effects, which show quantitative value in our experiments in Sec. 3.2 .
>
> **Combinations:**
> In some screens combinations of treatments exist, i.e. multiple CRISPR interventions changing the genetics, or multiple chemicals having been applied to a cell.
>
> **722 perturbations:**
> Thank you for catching this. There are indeed 722 perturbations and we train the model with all of them. However, when we made this visualization we dropped unannotated treatments to make it more legible and easier to annotate/sort by pathway. Our quantitative results on IWELBO contain results over the full set, we just made the figure more compact for visualization purposes and for the pathway prediction task which is not testable for treatments without annotations. We will of course explain and clarify this better and apologize for the confusion. We assure you that no ill intent or hidden performance problems exist with this discrepancy, it is simply omitted in the figure explanation.
> We are sharing the composition over the 722 here, showing that we filtered the 377 with label “none”. These labels are obtained from the replogle dataset preprocessing accompanying the SVAE+ paper.
>
> None                          377;
> S40_RIBOSOMAL_UNIT            101;
> S60_RIBOSOMAL_UNIT             56;
> S39_RIBOSOMAL_UNIT             44;
> SPLICEOSOME                    38;
> MT_PROTEIN_TRANSLOCATION       37;
> MEDIATOR_COMPLEX               26;
> NUCLEOTIDE_EXCISION_REPAIR     23;
> EXOSOME                        20;
>
> **Mask application and assumptions:**
> The mask is applied as the model shows element-wise to the treatment variable that is an inferred embedding before that is added to the basal state variable to model a cell.
> The model takes the shape z = e_t * m_t + z^b , where each of these objects has the same dimensionality of the embedding space D_z. As such our model determines through m_t for each treatment which dimension of the cell embedding gets shifted and using e_t for each one that gets shifted separately by how much. This allows us to model natural variation in an object that does not depend on a perturbation separately in the unperturbed dimensions, while also allowing for multiple different mechanisms to exist in superposition since each one has its own index and can be applied.
> We hope this also explains the additive nature of the model better: if these mechanisms are independent of each other we can apply them separately and we observe that our model outperforms every baseline in the combinatorial modeling task with that inductive bias, in particular the SVAE+ baseline which is the most recent paper, while the CPA-VAE version we compare to is imbued with our own inferential advances to perform better than its baseline version would and still underperforms the sparsified SAMS-VAE slightly.
>
>
> **Title too broad:**
> We agree that we primarily consider perturbations on cells in this manuscript and a simulation, since this is a lucid scenario where we have data of the form p(y|d). We could model images, proteomics, sequencing, any variable imaginable for this problem with our proposed model.
> If the reviewer is aware of any other scenario with public data of this very general structure we’d be happy to account for it. We want to note that the structure we propose could also be used for RL purposes with disentanglement over actions/interventions, but we do not aim to perform such a variant any time soon ourselves. As such, we’d be happy to adjust the title to reflect the focus on biology since this is a domain where reasoning over interventions makes sense, but we also want to note that our model has sufficiently general structure for a category of problems, way outside genomics.
>
> **IWELBO equation going over margins:**
> Thank you for catching this, we are adjusting this with a linebreak and apologize for the visual disturbance, we fully agree this is aesthetically unnecessary and suboptimal.

---

> > ### Comment · Reviewer_y8vh · 2023-08-16
> > **Respond to the Rebuttal**
> >
> > Thank you for your response.
> >
> > **Disentanglement**: Thank you for the clarifications. I understand the contributions but this is why I also mentioned that the title can miss-lead people. The title contains "...For Disentangled Representation Learning" therefore, and as there are different notions of disentanglement, you want to make it clear that the your notion is closer to causal discovery.
> >
> > **Figure 2: I understand the additive choice a bit better now. Thank you for your explanation regarding Figure 2. I am happy to take your word for it.

---

### Official Review · Reviewer_9yRM · 2023-07-24

**Soundness:** 3 good
**Presentation:** 2 fair
**Contribution:** 3 good
**Rating:** 6
**Confidence:** 4

**Summary:**

This paper proposes a new method, SAMS-VAE for learning disentangled representations leveraging the sparsity mechanism assumption. SAMS-VAE is bridging the gap between two existing models - CPA and SVAE+, hence resembles both. In simulated and real (single-cell) data experiments, the authors claim improvement over baselines with regards to ood prediction and interpretability of the latent space.

**Strengths:**

- Originality: SAMS-VAE fills the gap between two AE-based models, CPA which includes learning basal state of a cell without specifying a likelihood model over the latents, and SVAE+ which has a learnable model over the latents but no flexibility for representing basal states. Having both, makes the contribution of SAMS-VAE quite nice.

- Quality: The paper includes both technical and empirical results which with (some exceptions, see below) seem solid.

- Clarity: The manuscript is well written, although dense at places and the reader could benefit from more details in the supplementary.

- Significance: SAMS-VAE can be a quite helpful, relevant tool for the genomics community.

**Weaknesses:**

1. The title reads too broad, the model and method proposed indeed targets single cell applications, and without further extensions, and elaborations I don't see it as a general method that can extend to other data types. The authors should clarify this.
2. It would be helpful if the authors included the derivation of Eq 4 , 5.
3. I may have missed it, but where are the results for significantly improved ability to recover factors predictive of known molecular pathways, as stated in the introduction.


**Questions:**

In addition to my notes stated in the section above, it would help if the authors can answer the following:
1. There is no UMAPs for the latent representation of the baselines, could you include that for supporting the statement about improved interpretability?
2. What is the configuration you use for the simulated study? [12] proposes different setups (latent dimensions, strength of causal effect, sparsity level), which one in particular did you use? Also, why are other metrics avoided, especially the disentanglement ones R2, MCC?
3. Are there any identifiability guarantees of your model? In particular, how does your model cover modeling a discrete case such as gene expression data.
4. Is comparison wrt MSE or absolute error possible for the OOD setup, IWELBO is a bit limiting for comparison in applications  and also model specific.

**Limitations:**

Yes.

---

> ### Author Rebuttal · Authors · 2023-08-10
>
> We thank the reviewer for their positive review and time spent on out paper.
>
> **Derivation of Eq.4,5:**
> Eqs 4 and 5 indicate choices of variational families we design to perform inference in the model. As such, they are proposed by us and form a key part of our contribution in modeling the posterior of both SAMS-VAE and CPA-VAE significantly better than a naive variational inference approach would do, as indicated by our results ablating them in Table 1. We ablate CPA-VAE to isolate the sparsity contribution from SAMS-VAE from our inference improvements.
> In CPA, SAMS-VAE, SVAE+, we have multiple variables interacting, and in previous works their approximate posteriors are handled as independent (i.e. mean field). We introduce increasing degrees of correlation in the approximate posterior (Eq. 4 being the simple mean field, 5 using correlated embeddings and correlated encoders, and an intermediate step we describe in the paragraph modeling q(e_t) without the dependence on m), showing that increasing their complexity yields significantly better results in modeling cells by allowing the different interacting variables to “coordinate”.
> Section 2.3 explains a big part of our ML contributions for these classes of models regarding how we perform inference more successfully.
>
> **Title:**
> We thank the author for the comment. While our method is applied to single cell sequencing and a simulation taken from prior ML literature (see Appendix), we agree that we could easily add more specificity to the title to show we model cell perturbations.
> We want to add that nothing prevents exactly this model from being applied also to microscopy images or any other readout of cells; We leave that to future work as the practicalities of engineering high dimensional readouts with a VAE require bespoke work in the decoder and likelihood for each modality here exceeding the scope of this paper, but the key mechanisms of SAMS-VAE would remain very much intact. If the authors have another idea for a common ML-Benchmarking system that corresponds to the setup of our model we’d be happy to describe it here, as well.
>
>
> **Results & recovery of molecular pathways:**
> In Table 1, we summarize quantitative results for our model. On the rightmost column, we compare the ability of the model to disentangle cell embeddings according to whether dimensions correspond to a perturbation effect or not (versus being natural variation of a cell independent of perturbation). We then train classifiers on subsets of these and predict pathways which exist in the annotation of the underlying data, but the model is unaware of. We perform this experiment for both SVAE+ by Lopez et al 2023, and for various ablations of SAMS-VAE (with different inference strategies). The results show that SVAE+ gets an accuracy of 0.69 vs 0.9 for SAMS-VAE, indicating that the disentanglement corresponds much tighter to abstractions of pathways in embedding space. We hope this experiment makes sense to the reviewer.
> One practical utility of being able to do this is that one can inspect which subspaces of variation (and corresponding decoded features when varying those dimensions) are for instance relevant to a particular perturbation, and when electing perturbations to correspond to disease-modulators, being able to drill down in more detail into involved mechanisms of action.
> This is the task commonly performed in target discovery when performing high throughput screens, whereby looking for “signatures” of variation is the key task, and our model aids the discovery of more specific signatures as subspaces in an embedding containing many different effects relevant to the state of a cell/system.
>
>
> **Identifiability Guarantees:**
> In previously published models there are only identifiability guarantees (i.e. in the SVAE paper) when the decoder is ensured to have specific Lipschitz-properties, they do not exist in full generality for this model class. The SVAE+ paper by Lopez et al. also does not provide them. This is why we explicitly go for the term “disentanglement” and not “causality” in our paper.
> We believe that showing identifiability (and causality) is enormously difficult in such a general model class as a general VAE (with nonlinearities) without significant caveats and limitations, and indeed has not been done in any comparable papers we are aware of as of submission of this work.
> We thus prioritize generalization and measure it via the loglikelihood in in-distribution data, out-of-distribution data, and by measuring recovery of pathways successfully compared to other models. Our results suggest that getting inference to work better gets us a long way towards better generalization and practical utility.
>
> **OOD MSE:**
> Our likelihood is over discrete count data, as commonly done in the probabilistic literature for genomic data following numerous papers. We are afraid that MSE is not compatible either with the preprocessing or assumptions in our data.
> We consciously and laboriously elected to use the same likelihood used in SVAE+ over all our baselines to make IWELBO fully comparable in Table 1, including a carefully built generative version of CPA to make it fully comparable.
> However, to make models more generally comparable, we also perform an analysis in the appendix capturing average treatment effect, which models correlations of the predictive distribution of our model given a perturbation compared to differential expression. Our models do well in that metric, too, but we want to point out that this metric is flawed since differential expression and indeed absolute error severely underestimate aleatoric uncertainty in the count data, which is a huge factor in the present counts, and the probabilistic models are more sensitive to.
>
> **Umaps:**
> We are happy to add UMAPs and visualizations for other relevant models in the appendix of an updated paper. We want to point out that we did add visualizations of the latent factors already in Figure 2.

---

### Author Rebuttal · Authors · 2023-08-10

We thank our 4 reviewers for their feedback and questions.

The reviewers agreed that the paper is sound, interesting, and mostly clear and well-written.
While we addressed many of the individual concerns, we also want to summarize some of the shared discussion.

(I) we received questions about how much stronger our results are compared to baselines.
(II) reviewers are worried about identifiability and causality and how we think about that.
(III) the semantics and biological relevance of the disentangling and out overall set-up are not clear to all the reviewers.

We hope to give a general response to all three that supplements your individual responses and that you will find your questions positively resolved.

**Our contributions:**
Quick summary of our contributions.
First, a minor contribution is that we propose a generative version of the CPA model and coin it CPA-VAE, imbuing it with a count-based likelihood to make it comparable to other works in this line and affording it a generative interpretation and evaluation.
Second, our key change is that we propose adding a disentangling mechanism to that model by adding a mask m_t (same dimension as the embeddings) per treatment variables which allows the model to perform variable selection for which dimensions are useful to perturb to model cells under perturbation p(y|d) where d indications a dose vector/indicator.
Third, in Sec 2.3 we propose 2 changes to the basic VAE inference machinery tailored to our models, by capturing correlations in the posterior between the treatment variable and the basal state (effectively allowing the model to explain away treatment influences relative to basal state); and correlations between the masks and the treatment variables, thus allowing the model to not spend capacity on "switching on"  perturbation for dimensions that do not carry relevant information.

**Performance:**
We apply our improved inference machinery also to CPA-VAE in our experiments, and still show that SAMS-VAE outperforms both CPA-VAE and SVAE+, its most recent comparable model in the literature, in terms of modeling cells in-distribution, out-of distribution generalization in combinatorial settings with multiple treatments, and pathway recovery tasks based on the quality of the disentanglement.
We note that CPA-VAE mean field is the fairest baseline here, since improving the variational family is a contribution of our work, and as such believe Table 2 and Fig 4 show conclusively improved performance compared to prior work across the board.
In addition we share a simulation task inspired by previous papers in the appendix which shows that the model improves its empirical ability to recover/identify latent variables as its likelihood improves with better inference, and an evaluation framework using average treatment effect that is model agnostic and can be used across generative and non-generative models for cellular perturbations, also showing our models to perform very well.

**Causality/identifiability:**
We do not make claims about causality, but we are encouraged about the fact that our model generalizes better than its comparators in-distribution, out-of-distribution, and in a quantitative interpretation task where we infer previously known pathways.
We cannot guarantee causality, but raw predictive performance across all those by making a model more structured sounds quite promising to us as an intermediate step that is useful to inform further scientific inquiry.
Our simulation also shows that the model improves identification of latents as its NLLK improves with increasing sample-sizes available for training.

**Biological Interpretation:**
There are works in Genomics which utilize knowledge of specific pathways etc. in the model to improve performance. We want to explain why we do not consider that in this work and focus on more general ML.
In the setting we consider, which is a screen for cell responses given perturbations, we oftentimes may use chemicals as inputs or CRISPr interventions, and in the case of chemicals we do not know a priori which mechanisms they affect. In fact, the goal of a screen is to perform an unbiased study to surface the mechanisms of action affected by perturbations. It is useful to have an unbiased approach beyond known pathways and genetic perturbations, permitting for chemical perturbations and the existence of unknown mechanisms, as is the case in most practical screens.
Further, while the transcriptomics readout we have is a practical one for evaluating a paper given the existence of baseline work and comparable benchmarks in public, many biology screens also use other readouts given the relative expense of single cell sequencing and its lack of scalability. For instance, a popular readout is cell imaging, where cells are taken images of and used as readouts. An example of this is the CPJump dataset ( https://jump-cellpainting.broadinstitute.org/ ), which perturbs cells at scale using chemistry and genetic perturbations and visualizes cell-painting using microscopy.
Our model would be directly applicable to such a scenario using modifications for images, which we leave as future work.
We wrote this paper anchored on the experiments we elected because there is comparable work in ML that we can harden the modeling side against to set up the machinery to be rigorous, but we very consciously keep the model structure more general to account for broader applicability to modern quantitative biology.


We believe this work captures the best of CPA and SVAE+, adds inference tricks and quantitative rigor, and the resulting model shows a lot of promise for being useful in a variety of tasks strongly corresponding to the needs in screening in modern quantitative biology, while also being of general use to the community by showing what tricks were needed to make it work and how we can disentangle VAEs in more structured ways than the previous papers published in ML along the lines of our work.

---

### Decision · Program_Chairs · 2023-09-21

**Decision:**

Accept (poster)

**Comment:**

Although this paper had overall borderline reviews initially, the author response and subsequent discussion period did well to clarify many issues, with one reviewer raising the score accordingly. Two other reviewers did not raise their scores, but argued strongly that this paper should be accepted during private discussion between reviewers and the AC.

As AC, I would note that the final revision of the paper should reflect the discussion with the reviewers during the rebuttal period, and include the proposed edits.